# RoCA: A Robust Method to Discover Causal or Anticausal Relation by Noise Injection

## Abstract

Understanding whether the data generative process is causal or anticausal is important for algorithm design. It helps machine learning practitioners understand whether semi-supervised learning should be employed for real-world learning tasks. In many cases, existing causal discovery methods cannot be adaptable to this task, as they struggle with scalability and are ill-suited for high-dimensional perceptual data such as images. In this paper, we propose a method that detects whether the data generative process is causal or anticausal. Our method is robust to label errors and is designed to handle both large-scale and high-dimensional datasets effectively. Both theoretical analyses and empirical results on a variety of datasets demonstrate the effectiveness of our proposed method in determining the causal or anticausal direction of the data generative process.

## 1 Introduction

In real-world machine learning applications, acquiring unlabeled data is usually an easy process. However, the task of annotating data can be both time-consuming and expensive, which often results in the availability of a small amount of annotated data (Castro et al., 2020; Sohn et al., 2020). Training deep neural networks on limited annotated data can hinder their generalization ability (Kawaguchi et al., 2017; Xie et al., 2020; Verma et al., 2022). This has motivated a great deal of work on semi-supervised learning (SSL) (Laine & Aila, 2016; Sohn et al., 2020; Englesson & Azizpour, 2021; Amini et al., 2022; Laine & Aila, 2016; Kingma et al., 2014; Harris & Mazibas, 2013; Blanchard et al., 2010), which seeks to improve model generalization by also leveraging unlabeled data.

Let $\boldsymbol{X}$ represent the variable for instances, and let $Y$ denote the class. Existing research (Chapelle et al., 2006; Schölkopf et al., 2012; Kügelgen et al., 2020) has shown that the success of SSL depends on whether the instance distribution $P(\boldsymbol{X})$ provides information about the class-posterior distribution $P(Y|\boldsymbol{X})$. The extent of this information is influenced by the causality inherent in the data generative process. In an *anticausal setting*, the class $Y$ influences some causal variables in $\boldsymbol{X}$ during the data generative process (Schölkopf et al., 2012). In this context, the instance distribution $P(\boldsymbol{X})$ contains information about the class-posterior distribution $P(Y|\boldsymbol{X})$. Under such conditions, certain prerequisites for SSL, such as the clustering condition (Kügelgen et al., 2020), low-density separation (Chapelle & Zien, 2005), and manifold condition (Niyogi, 2013), are likely to be met. As a result, SSL can enhance the model's generalization capability. Conversely, in a *causal setting*, where the class $Y$ doesn't influence any causal variables in $\boldsymbol{X}$, but some causal variables in $\boldsymbol{X}$ influence $Y$, the distribution $P(\boldsymbol{X})$ doesn't offer insights about $P(Y|\boldsymbol{X})$ (Schölkopf et al., 2012). In such scenarios, SSL typically fails to enhance the model's generalization ability.

Understanding the causal or anticausal setting of a classification dataset is crucial for machine learning practitioners to determine the applicability of SSL. However, the setting is often unknown in real-world datasets and requires inference. Directly applying existing causal discovery methods (Kalainathan et al., 2020; Shimizu et al., 2011; Huang et al., 2018; Geiger & Heckerman, 1994; Zhang & Hyvarinen, 2009; Peters et al., 2011; 2014; Chen & Chan, 2013) to determine this tends to be impractical. One challenge is the difficulty in scaling. When faced with a multitude of causal variables, the computational demands escalate. Many existing causal discovery methods demand pairwise conditional independence tests for every potential edge between two variables (Akbari et al., 2022). This can lead to an exponential surge in runtime, especially when there's no prior knowledge to guide the discovery process (Le et al., 2016). While score-based greedy algorithms offer a potential

solution by enhancing computational efficiency, they come with their own set of challenges. Due to their inherent greedy search approach, these algorithms can get trapped in local optima, leading to potential inaccuracies in determining the causal or anticausal nature of a dataset (Glymour et al., 2019b). Another challenge arises when dealing with datasets that consist of perceptual data, such as images or audio. In these situations, causal variables remain unobservable (Schölkopf et al., 2021). Most existing causal discovery methods are tailored to detect relationships between observed causal variables, making them ill-suited for these types of datasets. As of now, we are unaware of any method that can effectively determine causal or anticausal relations in such datasets.

Adding to the aforementioned challenges, in real-world scenarios, observed labels in large-scale datasets can contain label errors (Deng et al., 2009; Xiao et al., 2015; Li et al., 2019) which have not been considered by existing causal discovery methods. In the mining process of large-scale datasets, inexpensive but imperfect annotation methods are wildly employed, for example, querying commercial search engines (Li et al., 2017), downloading social media images with tags (Mahajan et al., 2018), or leveraging machine-generated labels (Kuznetsova et al., 2020). These methods inevitably yield examples with label errors. When label errors are present, the randomness of these errors affects the strength of the causal association between features and the observed (noisy) label $\tilde{Y}$, making it more challenging to accurately discern the true relationships. For instance, existing methods employ conditional independence tests (Zhang & Hyvarinen, 2009; Peters et al., 2011; 2014) or score optimizations (Imoto et al., 2002; Hyvärinen & Smith, 2013; Huang et al., 2018) to evaluate the strength and structure of these relationships. Label errors introduce random fluctuations that distort the underlying relationships between features and labels. Consequently, these tests or score optimizations may be misled by the noise, leading to inaccurate estimations of the causal associations.

In this paper, we introduce a robust method aimed at determining whether a dataset is causal or anticausal. Recognizing that our primary interest lies in the causal associations between features and labels to assess whether the instance distribution $P(\boldsymbol{X})$ is informative to the class-posterior distribution $P(Y|\boldsymbol{X})$, we can focus on these relationships rather than all possible causal associations among all causal variables with common causality assumptions, i.e., faithfulness and acyclic graph assumption (Pearl, 2000). Consequently, the extensive computational costs and strict assumptions required for recovering or identifying all potential causal associations among one-dimensional variables are not necessary. However, when data contains label errors, verifying whether the distribution of instances $P(\boldsymbol{X})$ carries relevant information about class-posterior distribution $P(Y|\boldsymbol{X})$ is difficult, as the clean class $Y$ is latent. We found that the noisy class-posterior distribution $P(\tilde{Y}|\boldsymbol{X})$ can be used as an effective surrogate for $P(Y|\boldsymbol{X})$. The intuition of using $P(\tilde{Y}|\boldsymbol{X})$ as a surrogate of $P(Y|\boldsymbol{X})$ is that these two distributions are generally correlated, then if $P(\boldsymbol{X})$ carries relevant information about $P(Y|\boldsymbol{X})$, it also carries relevant information about $P(\tilde{Y}|\boldsymbol{X})$.

The core idea of our method is to check if the distribution of instances $P(\boldsymbol{X})$ carries relevant information about the prediction task $P(\tilde{Y}|\boldsymbol{X})$ to determine whether a dataset is causal or anticausal. To achieve it, we generate clusters by employing advanced unsupervised or self-supervised methods (Van Gansbeke et al., 2020; Ghosh & Lan, 2021). Then a pseudo label $Y'$ is assigned to each cluster based on the majority of observed labels within the cluster. To identify regions that can help predict observed (noisy) labels, different levels of noise are manually injected into observed labels, and the correlation of mismatch (disagreement) between pseudo labels and observed labels after noise injection is observed. In Section 3.3, we prove that in a causal setting, the mismatch and noise levels are not correlated; in an anticausal setting, the mismatch and noise levels are correlated. Experimental results on synthetic and real datasets demonstrate that our method can accurately determine the causal or anticausal direction.

It is worth noting that the application of our method extends beyond merely determining if a dataset is causal or anticausal. It can also be used to detect the causal direction between a set of continuous (or discrete) variables and a discrete variable. Specifically, there are cases where, based on prior knowledge or existing causal discovery methods, one recognizes the existence of causal associations between the variable set and a discrete variable, but the direction of this causality remains elusive. In such scenarios, our method is applicable.

## 2 PRELIMINARIES

Let $D$ be the distribution of a pair of random variables $(\boldsymbol{X}, \tilde{Y}) \in \mathcal{X} \times \{1, \ldots, C\}$, where $C$ denotes the number of classes, $\boldsymbol{X} \in \mathbb{R}^d$ represents an instance, and $\tilde{Y}$ denotes observed label which may not be identical to the clean class $Y$. Given a training sample $\boldsymbol{S} = \{\boldsymbol{x}_i, \tilde{y}_i\}_{i=1}^m$, we aim to reveal whether the dataset is a causal or an anticausal. Owing to space limitations, a review of existing causal discovery methods is left in Appendix B.

**The Principle of Independent Mechanisms** According to independent mechanisms (Peters et al., 2017b), the causal generative process of a system's variables consists of autonomous modules. Crucially, these modules do not inform or influence each other. In the probabilistic cases (detailed in Chapter 2 of Peters et al. (2017b)), the principle states that "*the conditional distribution of each variable given its causes (i.e., its mechanism) does not inform or influence the other conditional distributions.*" In other words, assuming all underlying causal variables are given and there are no latent variables, the conditional distributions of each variable, given all its causal parents (which can be an empty set), do not share any information and are independent of each other. To explain the independence concretely, we include an example in Appendix D. Note that in the case of two variables, a cause variable $C$ and an effect variable $E$, the principle simplifies to the independence between the cause distribution $P(C)$ and the effect distribution $P(E|C)$ (Schölkopf et al., 2012).

**Causal or Anticausal** We follow the definition of Causal and Anticausal datasets from Schölkopf et al. (2012). For causal datasets, *some variables in $\boldsymbol{X}$ act as causes for the class $Y$, and no variable in $\boldsymbol{X}$ is an effect of the class $Y$ or shares a common cause with the class $Y$* (e.g., Fig. 1a). In this case, $Y$ can only be an effect of some variables in $\boldsymbol{X}$. Two distributions $P(\boldsymbol{X})$ and $P(Y|\boldsymbol{X})$ satisfy the independent causal mechanisms. The distribution $P(\boldsymbol{X})$ does not contain information about $P(Y|\boldsymbol{X})$.

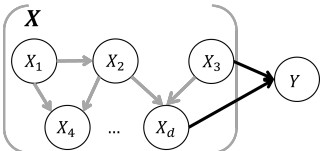

(a) The causal setting.

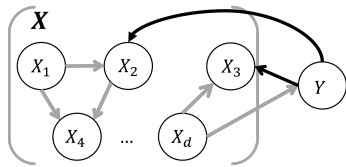

For anticausal datasets, however, the label $Y$ can be a cause variable. In such cases, the independent causal mechanisms are not satisfied for $P(\boldsymbol{X})$ and $P(Y|\boldsymbol{X})$, implying that $P(\boldsymbol{X})$ contains information about $P(Y|\boldsymbol{X})$.

(b) The anticausal setting.

We assume that there are no latent confounders, similar to many existing causal discovery methods. If latent confounders exist, our method will interpret it as anticausal, as in such cases, $P(\boldsymbol{X})$ also contains information about $P(Y|\boldsymbol{X})$, resembling an anticausal case. To further check whether it is an anticausal or confounded case, existing methods specifically designed for finding latent confounders can be applied (Chen et al., 2022; Huang et al., 2022).

Figure 1: Examples of causal and anticausal settings. The direction of the back-colored edge determines whether a dataset is causal or anticausal.

## 3 A ROBUST CAUSAL AND ANTICAUSAL (*RoCA*) ESTIMATOR

In this section, we present RoCA, a practical and robust method designed to infer whether a dataset is causal or anticausal while taking into account the presence of label errors in observed labels.

### 3.1 RATIONALE BEHIND ROCA

We first introduce a generative process of a dataset, which may include label errors. Then we explain that even if label errors exist in observed labels, they can still be used to infer if a dataset is causal or anticausal. Lastly, we discuss how our robust method is designed by utilizing observed labels.

**Data Generative Processes with Label Errors** A dataset with label errors can be viewed as a result of a random process where labels are flipped based on certain probabilities. Data generation involves two stages (see Fig. 2). Initially, an annotator is trained using a clean set $Z$, acquiring specific prior knowledge, $\theta$, for the labeling task. This knowledge helps the annotator form an annotation mechanism $P_\theta(\tilde{Y}|\boldsymbol{X})$, approximating the true class posterior $P(Y|\boldsymbol{X})$. This mechanism, being correlated with $P(Y|\boldsymbol{X})$, provides insights into the true class posterior. In the annotation phase, the annotator encounters a new instance $\boldsymbol{X}$ without an observed clean class $Y$. Using the

prior knowledge $\theta$, the annotator assigns an observed label $\tilde{Y}$ based on $P_\theta(\tilde{Y}|\boldsymbol{X})$. This process can sometimes lead to mislabeling. It's noteworthy that $P_\theta(\tilde{Y}|\boldsymbol{X})$ generally maintain a dependence with $P_\theta(Y|\boldsymbol{X})$. Imagine if this dependence did not exist; the annotation mechanism $P_\theta(\tilde{Y}|\boldsymbol{X})$ would essentially be a random guess of $P(Y|\boldsymbol{X})$, rendering the observed label $\tilde{Y}$ meaningless. We will demonstrate that, due to this dependence, $P_\theta(\tilde{Y}|\boldsymbol{X})$ can serve as a surrogate for $P(Y|\boldsymbol{X})$ to help determine whether a dataset is causal or anticausal.

$P_\theta(\tilde{Y}|\boldsymbol{X})$ **as a Surrogate of** $P(Y|\boldsymbol{X})$    Following the principles of independent mechanisms, for causal datasets, $P(\boldsymbol{X})$ does not provide any information about $P(Y|\boldsymbol{X})$. When $Y$ is the cause of $\boldsymbol{X}$, $P(\boldsymbol{X})$ generally contains information about $P(Y|\boldsymbol{X})$. (Kügelgen et al., 2020; Peters et al., 2017b). Therefore, to determine whether a dataset is causal or anticausal, one can examine whether $P(\boldsymbol{X})$ can inform $P(Y|\boldsymbol{X})$. However, this requires both $P(\boldsymbol{X})$ and $P(Y|\boldsymbol{X})$ can be accurately estimated. When data contains label errors, the clean label $Y$ is latent, estimating $P(Y|\boldsymbol{X})$ challenging. One natural thought is to identify a surrogate distribution that can assist in determining the causal direction.

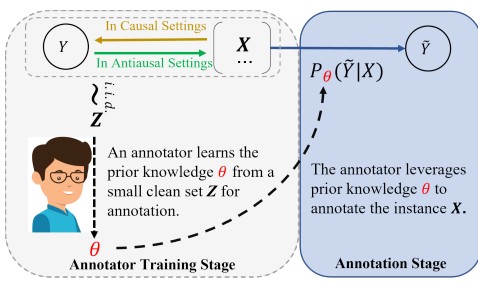

Figure 2: An illustration of annotation involving label errors.

In particular, the surrogate distribution must satisfy two key conditions. The first condition is that *under a causal setting, $P(\boldsymbol{X})$ should not be informative to the surrogate distribution.* The second condition is that *under an anticausal setting, $P(\boldsymbol{X})$ should be informative to the surrogate.* If such a surrogate distribution can be found, we can infer whether a dataset is causal or anticausal by examining whether $P(\boldsymbol{X})$ is informative to the surrogate distribution. To meet both conditions, our aim is to find a surrogate distribution that carries information about $P(Y|\boldsymbol{X})$ but remains disentangled with $P(\boldsymbol{X})$ under a causal setting. We find that $P_\theta(\tilde{Y}|\boldsymbol{X})$ fits these requirements. As it estimates the underlying distribution $P(Y|\boldsymbol{X})$. It is usually highly correlated with and informative about $P(Y|\boldsymbol{X})$. Moreover, under a causal setting, $P(\boldsymbol{X})$ cannot inform $P_\theta(\tilde{Y}|\boldsymbol{X})$, since $\tilde{Y}$ and $Y$ are effects of $\boldsymbol{X}$, and $P(\boldsymbol{X})$ and $P(\tilde{Y}|\boldsymbol{X})$ follows causal factorization and are disentangled according to independent mechanisms (Peters et al., 2017b). Thus, $P_\theta(\tilde{Y}|\boldsymbol{X})$ is a proper surrogate.

**Validating if** $P(\boldsymbol{X})$ **Informs** $P_\theta(\tilde{Y}|\boldsymbol{X})$    According to the above analysis that $P_\theta(\tilde{Y}|\boldsymbol{X})$ can be employed as a surrogate of $P(Y|\boldsymbol{X})$, the problem we need to solve is to effectively infer whether $P(\boldsymbol{X})$ informs $P_\theta(\tilde{Y}|\boldsymbol{X})$. To achieve it, our proposed method involves the use of clustering algorithms on $P(\boldsymbol{X})$ to generate clusters. We then assign a pseudo label $Y'$ to each cluster based on the majority of observed labels within it. If these pseudo labels are informative to observed labels, it indicates that $P(\boldsymbol{X})$ informs $P_\theta(\tilde{Y}|\boldsymbol{X})$. To check if these pseudo labels are informative to observed labels, we need to validate whether each pseudo label $Y'$ is a random guess of its corresponding observed label $\tilde{Y}$ given an instance $\boldsymbol{X}$. In other words, *we have to verify whether $P(\tilde{Y} = \tilde{y}|Y' = y', \boldsymbol{X} = \boldsymbol{x})$ equals $1/C$ for each instance $\boldsymbol{x}$.*

However, accurately estimating the distribution $P(\tilde{Y}|Y', \boldsymbol{X})$ from data can be difficult in general. Firstly, the instance or feature vector $\boldsymbol{X}$ can be high-dimensional, estimating the conditional probability distribution a daunting task because of the "curse of dimensionality" (Köppen, 2000). As the dimensionality increases, the data becomes sparse, and we require an exponentially larger amount of data to maintain the same level of statistical significance. More importantly, the distribution $P(\tilde{Y}|Y', \boldsymbol{X})$ can be diverse across different datasets. The diversity poses a significant challenge when trying to develop a robust and unified model that can accurately estimate the different distribution $P(\tilde{Y}|Y', \boldsymbol{X})$ across different datasets.

**Avoiding Estimation of** $P(\tilde{Y}|Y', \boldsymbol{X})$ **via Noise Injection**    To void directly estimating $P(\tilde{Y}|Y', \boldsymbol{X})$, we propose a simple and effective noise-injection method. We propose that we can inject different levels of instance-dependent noise to observed label $\tilde{Y}$, then compare the trend of the average disagreement between pseudo labels and modified labels under different levels of noise. The rationale is that, under a causal setting, $P(\boldsymbol{X})$ is not informative to both $P(Y|\boldsymbol{X})$ and $P(\tilde{Y}|\boldsymbol{X})$. Then

Figure 3: A illustration of our noise injection on causal and anticausal dataset.

exploiting $P(\boldsymbol{X})$ does not help predict observed labels. Therefore the pseudo labels obtained from $P(\boldsymbol{X})$ are random guesses of the observed labels. If we introduce noise to these observed labels by randomly flipping some of them, the pseudo labels should continue to guess the modified labels randomly. This is due to the fact that pseudo labels guess any label with a fixed probability of $1/C$. As such, the average disagreement between pseudo labels and the modified labels remains consistent, regardless of the noise level. As a result, we would not expect any trend in the average disagreement between pseudo labels and modified labels under different noise levels.

By contrast, under an anticausal setting, $P(\boldsymbol{X})$ is informative to both $P(Y|\boldsymbol{X})$ and $P(\tilde{Y}|\boldsymbol{X})$. This means that pseudo labels obtained by sufficiently exploiting $P(\boldsymbol{X})$ should not be simply random guesses of the observed labels. When we modify observed labels by injecting noise, these modified labels become increasingly random and unpredictable. This would result in a change of disagreement between the pseudo labels and the modified labels. As a result, we can expect a trend in the average disagreement between pseudo labels and modified labels under different noise levels.

**An Intuitive Illustration of RoCA**    Let's consider an example illustrated in Fig. 3. We're dealing with a binary classification dataset, where features $\boldsymbol{X} \in \mathbb{R}^2$. Assume that a clustering method separates instances into two clusters, with half of them assigned the pseudo label $Y' = 0$, and the other half assigned $Y' = 1$. We'll focus on instances with the pseudo label $Y' = 1$, which are located in two regions ($R_1$ and $R_2$) based on their $\boldsymbol{X}$ values.

In Fig. 3 (a), on a causal dataset, before noise injection, the distribution of observed labels in regions $R_1$ and $R_2$ indicate that $P(\tilde{Y} = 1|Y' = 0, \boldsymbol{X} = x) = P(\tilde{Y} = 0|Y' = 0, \boldsymbol{X} = x) = 1/2$. This suggests that each instance's pseudo label is a random guess of its observed label, rendering an average disagreement $P(\tilde{Y}^\rho = 1|Y' = 0)$ as $1/2$. after noise injection, say with an instance-dependent noise flipping $40\%$ and $20\%$ observed labels in regions $R_1$ and $R_2$, the average disagreement remains unaltered. It indicates no discernible trend in average disagreements between pseudo labels and modified labels across different noise levels.

Fig. 3 (b) demonstrates an anticausal dataset scenario. Despite the average disagreement for class $\tilde{Y} = 0$ being 0.5, each instance's pseudo label isn't a random guess of its observed label. In region $R_1$, all instances have the observed label $\tilde{Y} = 0$; while in region $R_2$, all instances have $\tilde{Y} = 1$. This results in $P(\tilde{Y} = 1|Y' = 0, \boldsymbol{X} = x) = 1$ in region $R_1$ and $P(\tilde{Y} = 0|Y' = 0, \boldsymbol{X} = x) = 1$ in region $R_2$, deviating from the expected $1/2$. After injecting the same instance-dependent noise into observed labels in regions $R_1$ and $R_2$, the average disagreement $P(\tilde{Y}^\rho = 1|Y' = 0)$ drops to 0.3, reflecting the regions where $P(\tilde{Y} = \tilde{y}|Y' = y', \boldsymbol{X} = x)$ doesn't equal $1/C$. Thus, our method successfully identifies this as an anticausal dataset.

### 3.2    Procedure of the RoCA Estimator

We employ a clustering method to exploit $P(\boldsymbol{X})$ and estimate pseudo labels $Y'$ for each cluster based on the majority of observed labels within it on training instances. Then we generate different sets of generated labels by manually injecting different levels of noise into the observed labels. By using 0-1 loss, we calculate the disagreement between pseudo labels and the generated labels with different injected noise levels, respectively. When $\boldsymbol{X}$ causes $Y$, the disagreement and the noise level should not be correlated in general. In contrast, when $Y$ causes $\boldsymbol{X}$, the disagreement (between $Y'$ and $\tilde{Y}$) and the noise level are statistically dependent.

**Learning Pseudo Labels**    The task of learning pseudo labels involves two steps: firstly, the data is clustered using a chosen clustering algorithm. Each cluster is then assigned a pseudo label, $Y'$, based on the majority of observed labels within that cluster. More specifically, consider $K = i$ as the $i$-th

cluster ID, and $\boldsymbol{X}_{k_i}$ as the set of instances with the $i$-th cluster ID, i.e.,

$$\boldsymbol{X}_{k_i} = \{\boldsymbol{x} | (\boldsymbol{x}, \tilde{y}) \in S, f(\boldsymbol{x}) = i\}, \tag{1}$$

where $f$ is a clustering algorithm that assigns an instance $\boldsymbol{x}$ with a cluster ID. Similarly, let $\boldsymbol{X}_{\tilde{Y}_j}$ denote the set of instances with the observed label $\tilde{Y} = j$. We define $\mathbb{1}_A$ as an indicator function that returns 1 if the event $A$ holds true and 0 otherwise. The pseudo label $Y'$ assigned to the instances in the set $\boldsymbol{X}_{k_i}$ is determined by the most frequent observed label within the cluster, i.e.,

$$Y' = \arg \max_{j \in C} \sum_{\boldsymbol{x} \in \boldsymbol{X}_{k_i}} \mathbb{1}_{\{\boldsymbol{x} \in \boldsymbol{X}_{\tilde{Y}_j}\}}. \tag{2}$$

Empirically, the assignment is implemented by applying Hungarian assignment algorithm (Jonker & Volgenant, 1986) which ensures an optimal assignment of pseudo labels to clusters such that the total number of mislabeled instances within each cluster is minimized.

**An Instance-Dependent Noise Injection** The core of our approach revolves around generating instance-dependent noise. We have to ensure that for any given instance $\boldsymbol{x}$, there's a dependence between its features and its flip rate $\rho_x$. As illustrated in Fig. 3, this dependence is pivotal for monitoring how the disagreement between pseudo labels and modified labels changes with different (average) noise levels in a dataset. Moreover, according to Theorem 1, in causal settings, to let average disagreement between pseudo labels and modified labels remain consistent across different noise levels, the noise must be introduced in a particular way. To be precise, for each flip rate $\rho_x$ of an instance $\boldsymbol{x}$, the probability of flipping an observed label to any other class should be uniformly distributed, which translates to $\frac{\rho_x}{C-1}$. The details are as follows.

To begin, for each instance in the dataset, we compute its $\ell_1$ norm using its features. These computed norms are stored in a vector $\boldsymbol{A}$. Subsequently, we generate a vector $\boldsymbol{P}$ of length $m$, where each element represents a distinct flip rate. These flip rates are derived from a truncated normal distribution $\psi$ with an average noise level $\rho$. The probability density function $\psi$ for this distribution is given by:

$$\psi(\mu = \rho, \sigma = 1, a = 0, b = 1; \rho_i) = \begin{cases} 0 & x \leq a \\ \frac{\phi(\mu, \sigma^2; \rho_i)}{\Phi(\mu, \sigma^2; b) - \Phi(\mu, \sigma^2; a)} & a < x < b \\ 0 & b \leq x. \end{cases}$$

Here, $\mu$ is the mean value, $\sigma$ is the standard deviation, $\phi$ and $\Phi$ are the probability density function and cumulative density function of a normal distribution, and $a$ and $b$ are the lower and upper limits of the truncated interval, respectively. To ensure that there's a dependency between the instances and the sampled flip rates in $\boldsymbol{P}$, we sort both $\boldsymbol{A}$ and $\boldsymbol{P}$ in ascending order. This step ensures that an instance with a smaller $a_i$ value corresponds to a lower individual flip rate $\rho_i$. The specific steps for our noise generation process are detailed in the pseudocode provided in Appendix F.

**Measuring Disagreement** To quantify the disagreement introduced by the noise, we adopted a non-parametric bootstrap approach where we resample the data $(X, \boldsymbol{Y})$ a large number of times (i.e., 30), and apply the noise injection procedure for each resampled dataset to get corresponding pseudo labels $\boldsymbol{Y}'$ and noisy labels $\boldsymbol{Y}^\rho$. Specifically, the modified label set $\tilde{\boldsymbol{Y}}^\rho$ is defined as the observed labels post the injection of our instance-dependent noise with an average noise level $\rho$. Note that these noise levels employed are randomly sampled from the range between 0 and 0.5.

The disagreement is then quantified using the 0-1 loss $\ell_{01}$ on the training examples. This is achieved by comparing $\boldsymbol{Y}'$ and $\tilde{\boldsymbol{Y}}^\rho$ with $0 - 1$ loss, i.e.,

$$\ell_{01}(\boldsymbol{Y}', \tilde{\boldsymbol{Y}}^\rho) = \frac{\sum_{i=1}^m \mathbb{1}_{\{y'_i \neq \tilde{y}_i^\rho\}}}{m}. \tag{3}$$

After measuring the disagreement. To infer whether a data set is causal or anticausal, the key lies in understanding the dependence between the disagreement and noise levels. In causal scenarios, there should not be any dependence, whereas in anticausal settings, a dependence is expected.

To achieve it, intuitively, we sample different noise levels and inject each noise level $\rho^i$ to observed labels and calculate the different average disagreement by using the 0-1 loss. A linear regression model is employed to characterize the dependences between noise level $\rho$ and the loss $\ell_{01}$, i.e.,

$$\{\hat{\beta}_0, \hat{\beta}_1\} = \arg \min_{\beta_0, \beta_1} \frac{1}{n} \sum_{i=1}^n (\ell_{01}^i - (\beta_1 \rho^i + \beta_0))^2, \tag{4}$$

where $\hat{\beta}_0$, $\hat{\beta}_1$ refer to the estimated intercept and slope of the regression line, $\ell_{01}^i$ denotes $0-1$ loss calculated under the $\rho^i$ noise level, respectively, and $n$ is the total number of noise levels. Accordingly, for causal datasets, the slope $\hat{\beta}_1$ should approximate $0$. In contrast, for anticausal datasets, this slope should deviate significantly from $0$.

## 3.3 THEORETICAL ANALYSES

In Theorem 1, we show that under the causal setting, the disagreement and the noise level should not correlated to each other, i.e., the slope $\beta_1$ is 0. In Theorem 2, we show that under the anticausal setting, the disagreement and the noise level are correlated to each other, i.e., the slope $\beta_1$ is not 0. It is important to note that our method is not only applicable in the special case when the observed labels contain instance-dependent label errors but also when they have no label errors or contain class-dependent noise.

Let $\mathcal{X}$ be the instance space and $C$ the set of all possible classes. Let $S = \{(x_i, \tilde{y}_i)\}_{t=0}^m$ be an sample set. Let $h : \mathcal{X} \to \{1, \ldots, C\}$, be a hypothesis that predicts pseudo labels of instances. Concretely, it can be a K-means algorithm together with the Hungarian algorithm which matches the cluster ID to the corresponding pseudo labels. Let $\mathcal{H}$ be the hypothesis space, where $h \in \mathcal{H}$. Let $\tilde{R}^\rho(h) = \mathbb{E}_{(\boldsymbol{x}, \tilde{y}^\rho) \sim P(\boldsymbol{X}, \tilde{Y}^\rho)}[\mathbb{1}_{\{h(\boldsymbol{x}) \neq \tilde{y}^\rho\}}]$ be the expected disagreement $\tilde{R}(h)$ between pseudo labels and generated labels $\tilde{y}^\rho$ with $\rho$-level noise injection. Let $\hat{\tilde{R}}_S^\rho(h)$ be the average disagreement (or empirical risk) of $h$ on the set $S$ after $\rho$-level noise injection. Theorem 1 and Theorem 2 leverage the concept of *empirical Rademacher complexity*, denoted as $\hat{\mathfrak{R}}_S(\mathcal{H})$ (Mohri et al., 2018).

**Theorem 1** (Invariant Disagreements Under the Causal Settings). *Under the causal setting, assume that for every instance and clean class pair $(x, y)$, its observed label $\tilde{y}$ is obtained by a noise rate $\rho_x$ such that $P(\tilde{Y} = \tilde{y}|Y = y, \boldsymbol{X} = x) = \frac{\rho_x}{C-1}$ for all $\tilde{y} \neq y \wedge \tilde{y} \in C$. Then after injecting noise to the sample with arbitrary average noise rates $\rho^1$ and $\rho^2$ such that $0 \leq \rho^1 \leq \rho^2 \leq 1$, with a $1 - \delta$ probability and $\delta > 0$,*

$$|\hat{\tilde{R}}_S^{\rho^1}(h) - \hat{\tilde{R}}_S^{\rho^2}(h)| \leq 4\hat{\mathfrak{R}}_S(\mathcal{H}) + 6\sqrt{\frac{\log \frac{4}{\delta}}{2m}}. \tag{5}$$

As the sample size $m$ increases, the term $3\sqrt{\frac{\log \frac{4}{\delta}}{2m}}$ tends towards 0 at a rate of $\mathcal{O}(\frac{1}{\sqrt{m}})$. Additionally, the empirical Rademacher complexity $\hat{\mathfrak{R}}S(\mathcal{H})$ of the K-means algorithm also tends towards 0 at a rate of $\mathcal{O}(\frac{1}{\sqrt{m}})$, as demonstrated by Li & Liu (2021). Consequently, the right-hand side of Inequality (5) converges to 0 at a rate of $\mathcal{O}(\frac{1}{\sqrt{m}})$. This implies that with an increasing sample size, the difference between the disagreements $\hat{\tilde{R}}_S^{\rho^1}(h)$ and $\hat{\tilde{R}}_S^{\rho^2}(h)$, obtained by introducing different noise levels, will tend towards 0. In other words, the level of disagreement remains unaffected by changes in noise levels, consequently leading to the conclusion that the slope $\beta_1$ equals zero.

**Theorem 2** (Variable Disagreements Under the Anticausal Setting). *Under the anticausal setting, after injecting noise with a noise level $\rho = \mathbb{E}_X[\rho_x]$, $\tilde{R}^\rho(h) - \tilde{R}(h) = \mathbb{E}\left[\left(1 - \frac{C\tilde{R}(h,x)}{C-1}\right)\rho_x\right]$.*

Theorem 2 shows that the difference of the risk after noise injection between the risk on observed labels is $\mathbb{E}\left[\left(1 - \frac{C\tilde{R}(h,x)}{C-1}\right)\rho_x\right]$. Under the anticausal setting, the pseudo labels predicted by $h$ are not random guesses. In this case, $\tilde{R}(h,x) \neq (C-1)/C$, then the difference is always nonzero. It implies that after injecting noise, the slope $\beta_1$ will be nonzero.

**Assumptions for Discovering Causal and Anticausal** Our method is based on commonly accepted assumptions in causal disovery: causal faithfulness, acyclic graph assumption, absence of latent confounders, and independent causal mechanisms (Peters et al., 2014). To ensure that the disagreements (or expected risks) under different noise levels remain constant in a causal setting when employing RoCA, we need an additional assumption to constrain the types of label errors in datasets. Specifically, this assumption posits that for every instance and clean class pair $(x, y)$, the observed label $\tilde{y}$ is derived with a noise rate $\rho_x$ such that $P(\tilde{Y} = \tilde{y}|Y = y, \boldsymbol{X} = x) = \frac{\rho_x}{C-1}$ for all $\tilde{y} \neq y \wedge \tilde{y} \in C$. This assumption can satisfy not only when data contains instance-dependent label

Figure 4: The average disagreement and its standard deviation under different noise rates for synthetic datasets: synCausal (left) and synAnticausal (right).

errors but also when there are no label errors or when data contains class-dependent errors (Patrini et al., 2017; Xia et al., 2019; Li et al., 2021).

Furthermore, to use $P_\theta(\tilde{Y}|\boldsymbol{X})$ as a proxy for $P(Y|\boldsymbol{X})$, we assume a dependence between $P_\theta(\tilde{Y}|\boldsymbol{X})$ and $P(Y|\boldsymbol{X})$. This assumption is typically valid, as the absence of such a dependence would imply that the annotation mechanism $P_\theta(\tilde{Y}|\boldsymbol{X})$ is merely a random guess of $P(Y|\boldsymbol{X})$, rendering the observed label $\tilde{Y}$ meaningless.

Additionally, the effectiveness of our method can be influenced by the choice of a backbone clustering method. Specifically, when dealing with an anticausal dataset, our approach relies on a clustering method that is capable enough to extract some information from $P(\boldsymbol{X})$ for predicting $P(Y|\boldsymbol{X})$, rather than merely making a completely random guess. Thanks to the recent successes of unsupervised and self-supervised methods, some methods (Niu et al., 2021) based on contrastive learning have even achieved competitive performance compared to supervised methods on benchmark image datasets such as CIFAR10 and MNIST.

## 4 EXPERIMENTS

Our code is implemented using PyTorch. To obtain the pseudo label $Y'$, we employ the K-means clustering method (Likas et al., 2003) for non-image datasets. For image datasets, specifically *CIFAR10* and *Clothing1M*, we use the SPICE* clustering method (Niu et al., 2021). All models are trained on Nvidia V100 GPUs. We evaluated our RoCA estimator across 18 datasets. This includes 2 synthetic datasets (*synCausal* and *synAnticausal*), 13 multi-variate real-world datasets, and 3 image datasets (*CIFAR10*, *CIFAR10N*, *Clothing1M*). Notably, *CIFAR10N* and *Clothing1M* contain real-world label errors and are large-scale (1M images on Clothing1M). Our approach was benchmarked against 9 causal discovery baseline algorithms. *Most experimental results and descriptions of baseline methods and datasets are left in Appendix E.*

**Labels Errors and Implementation** Different label errors are employed to validate the robustness of RoCA estimator. (1) Symmetry Flipping (Sym) (Patrini et al., 2017) which randomly replaces a percentage of labels in the training data with all possible labels. (2) Pair Flipping (Pair) (Han et al., 2018) where labels are only replaced by similar classes. For datasets with binary class labels, Sym and Pair noises are identical. (3) Instance-Dependent Label Error (IDN) (Xia et al., 2020) where different instances have different transition matrices depending on parts of instances. To simulate scenarios where datasets contain label errors, the different errors are injected into the clean classes.

To rigorously validate the disagreement, rather than directly evaluating if the slope $\hat{\beta}_1$ contained from Eq. (4) is near 0, we use a hypothesis test on the slope. The level of noise $\rho$ is randomly sampled 20 times from a range between 0 and 0.5. For every selected noise level, a disagreement between $Y'$ and $\tilde{Y}$ can be calculated. Consequently, a slope value is calculated from the correlation of these 20 noise levels and their respective disagreement ratios. By repeating such procedure for 30 times, a set of slope values can be obtained. These slope values are then utilized in our hypothesis test to verify if the average slope is significantly different from 0. The details of this test are provided in Appendix A. Intuitively, if the resulting $p$-value from the test exceeds 0.05, the slope is likely close to zero, indicating a causal dataset. Conversely, a $p$-value below 0.05 suggests an anticausal dataset.

**Disagreements with Different Noise Levels on Synthetic Datasets** Fig. 4 demonstrates the trend of disagreement with different noise levels for *synCausal* and *synAnticausal* datasets. To construct datasets with label errors, 30% label errors are added into these datasets. For the *synCausal* dataset, the trend of disagreement remains unchanged at 0.5 with the increase of injected noise rates, and the

Table 1: Comparing with other baselines on UCI datasets.

| | Method | Original | Instance | | | Pair | | | Sym | | |
|---|---|---|---|---|---|---|---|---|---|---|---|
| | | 0% | 10% | 20% | 30% | 10% | 20% | 30% | 10% | 20% | 30% |
| KrKp (causal) | GES | anticausal | anticausal | anticausal | anticausal | anticausal | anticausal | anticausal | anticausal | anticausal | anticausal |
| | GIES | anticausal | anticausal | anticausal | anticausal | anticausal | anticausal | anticausal | anticausal | anticausal | anticausal |
| | PC | **causal** | anticausal | anticausal | anticausal | **causal** | anticausal | anticausal | **causal** | anticausal | anticausal |
| | ICD | **causal** | anticausal | anticausal | anticausal | anticausal | anticausal | anticausal | anticausal | anticausal | anticausal |
| | RAI | anticausal | anticausal | anticausal | anticausal | anticausal | anticausal | anticausal | anticausal | anticausal | anticausal |
| | FCI | anticausal | anticausal | anticausal | anticausal | anticausal | anticausal | anticausal | anticausal | anticausal | anticausal |
| | LINGAM | unknown | unknown | unknown | unknown | unknown | unknown | unknown | unknown | unknown | unknown |
| | SAM | unknown | unknown | unknown | unknown | unknown | unknown | unknown | unknown | unknown | unknown |
| | CCDR | **causal** | **causal** | unknown | unknown | **causal** | unknown | unknown | **causal** | unknown | unknown |
| | Our method | p=0.3263 **causal** | p=0.7719 **causal** | p=0.6757 **causal** | p=0.2009 **causal** | p=0.4315 **causal** | p=0.1548 **causal** | p=0.3520 **causal** | p=0.4315 **causal** | p=0.1564 **causal** | p=0.3504 **causal** |
| Splice (causal) | GES | anticausal | anticausal | anticausal | anticausal | anticausal | anticausal | anticausal | anticausal | anticausal | anticausal |
| | GIES | anticausal | anticausal | anticausal | anticausal | anticausal | anticausal | anticausal | anticausal | anticausal | anticausal |
| | PC | anticausal | anticausal | anticausal | anticausal | anticausal | anticausal | anticausal | anticausal | anticausal | anticausal |
| | ICD | anticausal | anticausal | anticausal | anticausal | anticausal | anticausal | anticausal | anticausal | anticausal | anticausal |
| | RAI | unknown | unknown | unknown | unknown | unknown | unknown | unknown | unknown | unknown | unknown |
| | FCI | anticausal | anticausal | anticausal | anticausal | anticausal | anticausal | anticausal | anticausal | anticausal | anticausal |
| | LINGAM | unknown | unknown | unknown | unknown | unknown | unknown | unknown | unknown | unknown | unknown |
| | SAM | unknown | unknown | unknown | unknown | unknown | unknown | unknown | unknown | unknown | unknown |
| | CCDR | anticausal | anticausal | anticausal | anticausal | anticausal | anticausal | anticausal | anticausal | anticausal | anticausal |
| | Our method | p=0.0022 anticausal | p=0.7749 **causal** | p=0.7748 **causal** | p=0.2731 **causal** | p=0.0395 anticausal | p=0.8958 **causal** | p=0.0000 anticausal | p=0.0085 anticausal | p=0.1976 **causal** | p=0.0314 anticausal |
| WDBC (anticausal) | GES | **anticausal** | causal | causal | causal | causal | causal | causal | causal | **anticausal** | causal |
| | GIES | **anticausal** | causal | causal | causal | causal | causal | causal | causal | **anticausal** | causal |
| | PC | **anticausal** | **anticausal** | **anticausal** | unknown | **anticausal** | unknown | unknown | **anticausal** | unknown | unknown |
| | ICD | **anticausal** | **anticausal** | **anticausal** | unknown | **anticausal** | **anticausal** | **anticausal** | **anticausal** | **anticausal** | **anticausal** |
| | RAI | unknown | unknown | unknown | unknown | unknown | unknown | unknown | unknown | unknown | unknown |
| | FCI | **anticausal** | **anticausal** | unknown | unknown | **anticausal** | unknown | unknown | **anticausal** | unknown | unknown |
| | LINGAM | unknown | unknown | unknown | unknown | unknown | unknown | unknown | unknown | unknown | unknown |
| | SAM | unknown | unknown | unknown | unknown | unknown | unknown | unknown | unknown | unknown | unknown |
| | CCDR | causal | causal | causal | causal | causal | causal | causal | causal | causal | causal |
| | Our method | p=0.0000 **anticausal** | p=0.0000 **anticausal** | p=0.0000 **anticausal** | p=0.0000 **anticausal** | p=0.0000 **anticausal** | p=0.0000 **anticausal** | p=0.0000 **anticausal** | p=0.0000 **anticausal** | p=0.0000 **anticausal** | p=0.0000 **anticausal** |
| Letter (anticausal) | GES | anticausal | anticausal | anticausal | anticausal | anticausal | anticausal | anticausal | anticausal | anticausal | anticausal |
| | GIES | anticausal | anticausal | anticausal | anticausal | anticausal | anticausal | anticausal | anticausal | anticausal | anticausal |
| | PC | anticausal | anticausal | anticausal | anticausal | anticausal | anticausal | anticausal | anticausal | anticausal | anticausal |
| | ICD | anticausal | anticausal | anticausal | anticausal | anticausal | anticausal | anticausal | anticausal | anticausal | anticausal |
| | RAI | unknown | unknown | unknown | unknown | unknown | unknown | unknown | unknown | unknown | unknown |
| | FCI | anticausal | anticausal | anticausal | anticausal | anticausal | anticausal | anticausal | anticausal | anticausal | anticausal |
| | LINGAM | unknown | unknown | unknown | unknown | unknown | unknown | unknown | unknown | unknown | unknown |
| | SAM | unknown | unknown | unknown | unknown | unknown | unknown | unknown | unknown | unknown | unknown |
| | CCDR | unkown | unkown | causal | causal | anticausal | anticausal | causal | **anticausal** | unknown | **anticausal** |
| | Our method | p=0.0000 **anticausal** | p=0.0000 **anticausal** | p=0.0000 **anticausal** | p=0.0000 **anticausal** | p=0.0000 **anticausal** | p=0.0000 **anticausal** | p=0.0000 **anticausal** | p=0.0000 **anticausal** | p=0.0000 **anticausal** | p=0.0000 **anticausal** |

Table 2: Performance of RoCA on large-scale image datasets containing label errors.

| Clothing1M | CIFAR10N | | | | | |
|---|---|---|---|---|---|---|
| | Clean | Worst | Aggre | Random1 | Random2 | Random3 |
| p=0.0000 **anticausal** | p=0.0000 **anticausal** | p=0.0000 **anticausal** | p=0.0000 **anticausal** | p=0.0000 **anticausal** | p=0.0000 **anticausal** | p=0.0000 **anticausal** |

slope $\hat{\beta}_1$ of the regression line is close to $0$. This is because $Y'$ is poorly estimated and should be a random guess of noised $\tilde{Y}'$. On the other hand, for the *synAnticausal* dataset with small label errors (e.g., Sym and Ins-10% to 20%), there is a strong positive correlation between the disagreement and the noise level. In this case, $Y'$ is better than a random guess of both $\tilde{Y}$ and the latent clean class $Y$. Specifically, with the increase of noise level $\rho$, the corresponding $\tilde{Y}^\rho$ becomes more seriously polluted and tends to deviate far away from the observed label $\tilde{Y}$. This results in a larger disagreement between $\tilde{Y}^\rho$ and $Y'$.

**Performance of RoCA on Real-World Datasets** We have also benchmarked the RoCA method against other causal discovery algorithms. Our results, as presented in Tab. 1 and Tab. 2, demonstrate that our method is both more accurate and robust. In these tables, the term 'unknown' indicates cases where the algorithm either failed to detect the causal relation, or did not complete the analysis within a feasible running time. Note that only RoCA can applied to image datasets CIFAR10N and Clothing1M to detect causal and anticausal relations.

## 5 CONCLUSION

This paper presents a label-error robust estimator for inferring whether a dataset is causal or anticausal. The intuition is to leverage the information asymmetry between the distributions $P(\boldsymbol{X})$ and $P(\tilde{Y}|\boldsymbol{X})$ of the observed label $\tilde{Y}$ on anticausal and causal datasets by noise injection. Our theoretical analyses and empirical results demonstrate the effectiveness of the RoCA estimator in determining the causal or anticausal nature of a dataset.

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

## A  THE HYPOTHESIS TEST

We perform the one-sample $t$-test to quantify whether the slope $\beta_1$ is significantly different from zero. To validate this, we perform a $t$-test, and the null hypothesis is that the slope $\hat{\beta}_1$ is zero. Let $T$ refer to the observed test statistic, $P_0$ denotes the t-distribution of the test statistic under the null hypothesis, then the $p$-valtue of the $t$-test where $\hat{\beta}_1$ is significantly different from zero are as follows.

$$p\text{-value } = \mathbb{P}\left(T \geq t^* \mid T \sim P_0\right), \quad t^* = \frac{\hat{\beta}_1 - 0}{\sqrt{\frac{1}{n-2}\frac{\sum_{i=1}^n \left(\ell_{01_i} - (\hat{\beta}_1 \rho_i + \hat{\beta}_0)\right)^2}{\sum_{i=1}^n (\rho_i - \bar{\rho})^2}}/\sqrt{n}}.$$

We check the condition of whether the larger $p$-value is less than the significance level $0.05$ or not. If the condition holds, the null hypotheses will be rejected, and the slope $\hat{\beta}_1$ is significantly different from zero. Then the dataset is anticausal. Otherwise, the null hypothesis cannot be rejected, suggesting the slope $\hat{\beta}_1$ is zero. Then the dataset is very likely to be causal.

## B A DETAILED REVIEW OF CAUSAL DISCOVERY METHODS

**Constraint-Based and Score-Based Approaches.** To build a graph that captures these conditional independencies, the majority of constraint-based techniques look for conditional independencies in the empirical joint distribution. Since numerous graphs frequently satisfy a given set of conditional dependencies, as was discussed above, constraint-based methods frequently produce a graph that represents some Markov equivalence classes. Unfortunately, large sample sizes are necessary for conditional independence tests to be reliable, and (Shah & Peters, 2020) highlights further difficulties in controlling Type I errors.

Score-based approaches test the validity of a candidate graph $\mathcal{G}$ according to some scoring function $S$. The goal is therefore stated as (Peters et al., 2017a):

$$\hat{\mathcal{G}} = \text{argmax}_{\mathcal{G} \text{ over } \mathbf{x}} S(\mathcal{D}, \mathcal{G}) \tag{6}$$

where the empirical data for the variables $\mathbf{X}$ is represented by $\mathcal{D}$. Common scoring functions include the Bayesian Information Criterion (BIC) (Geiger & Heckerman, 1994), the Minimum Description Length (as an approximation of Kolmogorov Complexity) (Janzing & Schölkopf, 2010; Grünwald & Vitányi, 2008; Kalainathan et al., 2020), the Bayesian Gaussian equivalent (BGe) score (Geiger & Heckerman, 1994), the Bayesian Dirichlet equivalence (BDe) score (Heckerman et al., 1995), the Bayesian Dirichlet equivalence uniform (BDeu) score (Heckerman et al., 1995), and others (Imoto et al., 2002; Hyvärinen & Smith, 2013; Huang et al., 2018).

**Functional Causal Models.** Methods based on causal function provide an alternate strategy for estimating causal effects. Assumptions about the data generation process are used in these causal function-based techniques. The causal function-based approach fits the causal function model among variables and then infers causal directions using causal assumptions, such as a non-Gaussian assumption of the noise (Shimizu et al., 2006; 2011) the independence assumption between cause variables and noise (Zhang & Hyvarinen, 2009; Peters et al., 2011; 2014) and the independence assumption between the distribution of cause variables and the causal function (Janzing et al., 2012). Most LiNGAM-based approaches for the linear case (Shimizu et al., 2006) assume non-Gaussian noise and linear causal relations between variables. This model seeks to determine a causal order among the random observed variables.

To deal with linear latent confounders, an estimation method utilizing overcomplete ICA (Lewicki & Sejnowski, 2000) is suggested. However, overcomplete ICA algorithms usually suffer from local optimum and cannot be employed when the number of variables is large.

By evaluating the independence between the estimated exogenous variables and the residual, (Tashiro et al., 2014) identify latent confounders. They discover that variables from subsets that are not impacted by latent confounders are included, and they estimate causal orders one at a time. (Chen & Chan, 2013) investigate linear non-Gaussian acyclic models in the presence of latent Gaussian confounders (LiNGAM-GC), which assumes that the latent confounders are Gaussian distributed independently.

## C CAUSAL GRAPHS AND STRUCTURAL CAUSAL MODELS (SCM)

Directed acyclic graphs (DAGs) serve as a formalism for representing causal relationships. In these graphs, arrows point from the parent node (direct cause) to the child node (direct effect) (Pearl, 2000). Building upon this graphical representation, a structural causal model (SCM) can be constructed to capture the causal mechanisms that underlie the data distribution.

An SCM is composed of a set of variables interconnected by functions, representing the flow of information. This model elucidates the causal relationships among variables, offering a detailed insight into the data generation process. Consider a DAG $G = (V, E)$ defined over a set

of variables $\{X_1, X_2, \cdots, X_d, Y\}$, with $P$ representing their joint distribution. Let $\boldsymbol{X}$ be the set $\{X_1, X_2, \cdots, X_d\}$. The notation $\boldsymbol{X}_{PA_i^G}$ refers to the direct causes of $X_i$, while $\boldsymbol{Y}_{PA^G}$ denotes the direct causes of $Y$. Disturbances or errors in the generative processes of $X_i$ and $Y$ are represented by $N_i$ and $N_y$, respectively. The SCM for a classification dataset can be expressed as:

$$X_i := f_i(\boldsymbol{X}_{PA_i^G}, N_i), \;\; i = 1, ..., d; \; Y := f_y(\boldsymbol{Y}_{PA^G}, N_y).$$

The causal factorization of the joint distribution is given by:

$$P(\boldsymbol{X}, Y) = P(Y|\boldsymbol{Y}_{PA^G}) \prod_i P(X_i|\boldsymbol{X}_{PA_i^G}). \tag{7}$$

It's worth noting that both $\boldsymbol{X}_{PA_i^G}$ and $\boldsymbol{Y}_{PA^G}$ are allowed to be empty sets.

## D  UNDERSTANDING THE INDEPENDENCE BETWEEN DISTRIBUTIONS

To concretely explain that the conditional distribution of each variable, given its causes (i.e., its mechanism), does not inform or influence the other conditional distributions, let's consider an interesting example that follows the generative process of causal datasets.

- We act as the data collector. 1). we randomly sample a photo $\boldsymbol{X}$ from Instagram.
- Let Tom be the annotator. He will annotate each $\boldsymbol{X}$ we pass but without any knowledge of $P(\boldsymbol{X})$.
- Following the generative process, 2). we pass the photo $\boldsymbol{X}$ to Tom. Tom writes the label $Y$ on the back of the photo $\boldsymbol{X}$ and puts the photo in a black box.
- We repeat the process 1), and Tom repeats the process 2).

The question then arises: *can we act like a clustering algorithm by looking at $P(\boldsymbol{X})$ to understand how photos in the box are labeled?* Generally, the answer is no. Intuitively, there are too many possible ways to annotate the photo. Tom could label the photos based on whether the image contains a human, the number of humans, night vs. day, and other characteristics. We have no idea about his mechanism by only looking at $P(\boldsymbol{X})$. In this case, $P(\boldsymbol{X})$ does not inform $P(Y|\boldsymbol{X})$.

## E  MORE EXPERIMENTS

### E.1  INTRODUCTION OF REAL-WORLD CAUSAL DATASETS

1. *KrKp* dataset contains 3196 instances with 36 attributes. Each instance is a board description for the chess endgame, where the feature attributes describe the board and the label determines whether it is "win" or "nowin". It is considered a causal dataset since the board description causally influences whether white will win.

2. *Splice* dataset contains 3190 instances with 60 attributes, where attributes describe sequential DNA nucleotide positions and the label is the type of splice sites. It is considered a causal dataset since the DNA sequence causes the splice sites.

3. *SecStr* dataset contains 83680 instances with 15 attributes, where attributes describe the amino acid and the label is the corresponding secondary chemical structure. It is considered a causal dataset since the secondary structure is determined by its amino acid features.

### E.2  INTRODUCTION OF REAL-WORLD ANTICAUSAL DATASETS

1. *WDBC* dataset contains 569 instances with 32 attributes. It is an anticausal dataset, where the class causes some of the tumor features.

2. *Letter* dataset contains 20000 instances with 16 attributes. It is an anticausal daset, where the class (letter) causes the produced image of the letter.

3. *Breastcancer* dataset contains 286 instances with 9 attributes. It is an anticausal dataset, where the class causes some of the tumor features.

4. *Coil* dataset contains 1500 instances with 241 attributes. It is considered an anticausal/confounded dataset because the six-state class and the features are confounded by the 24-state variable of all objects.

5. *G241C* dataset contains 1500 instances with 241 attributes. It is considered an anticausal dataset since the class determines the features.

6. *Iris* dataset contains 150 instances with 4 attributes. It is an anticausal dataset, where the size of the plant is an effect of the category.

7. *Mushroom* dataset contains 8124 instances with 22 attributes. It is an anticausal dataset, where the attributes of the mushroom and the class are confounded by the mushroom taxonomy.

8. *Segment* dataset contains 2310 instances with 19 attributes. It is an anticausal dataset, where the class causes the features of the image.

9. *Usps* dataset contains 1500 instances with 240 attributes. It is an anticausal dataset, where the class and the features are confounded by the 10-state variable of all digits.

10. *Waveform* dataset contains 5000 instances with 21 attributes attributes and 1 label. Each class is generated from a combination of 2 or 3 "base" waves. It is considered an anticausal dataset since the class of the wave causes its attributes.

11. *CIFAR10* dataset contains 60000 $32 \times 32$ color images (attributes) in 10 classes (label), with 6000 images per class. It is considered an anticausal dataset since the images are collected according to the predefined 10 different labels.

12. *CIFAR10N* has the same number of instances and attributes as those of *CIFAR10* while there are 5 different types of human-annotated real-world noisy labels from Amazon Mechanical Turk.

13. *Clothing1M* contains 1M clothing images in 14 classes. It is a causal dataset with noisy labels since the image determines its class and the data is collected from several online shopping websites.

14. *Digit1* dataset contains 1500 instances with 241 attributes. It is considered an anticausal dataset because the positive or negative angle and the features are confounded by the variable of continuous angle.

### E.3 INTRODUCTION OF SYNTHETIC DATASETS

In addition, we have generated two additional synthetic datasets, namely *synCausal* and *synAnticausal*, to validate our RoCA estimator. Each dataset consists of 20000 instances with 5 attributes and 1 label. In the case of *synCausal*, we generate each instance by randomly sampling 5 values from a standard normal distribution to represent $X$, and then compute the corresponding $y$ value using a polynomial function. This process simulates the data generative process where $X$ causes $Y$. Conversely, the instances in *synAnticausal* are generated similarly, but in the opposite direction, to reflect that $Y$ causes $X$.

### E.4 INTRODUCTION OF BASELINE CAUSAL DISCOVERY METHODS

The baseline causal discovery methods we employed are as follows.

1. GES (Chickering, 2002): The Greedy Equivalence Search algorithm is a score-based Bayesian approach that heuristically searches for a graph that minimizes a likelihood score on the given data.

2. GIES (Hauser & Bühlmann, 2012): The Greedy Interventional Equivalence Search algorithm is similar to GES, but it incorporates interventional data for inference.

3. PC (Spirtes et al., 2000b): The Peter-Clark algorithm is one of the renowned score-based methods for causal discovery. It efficiently employs conditional tests on variables and variable sets.

4. ICD (Rohekar et al., 2021): Iterative Causal Discovery recovers causal graphs in the presence of latent confounders and selection bias. ICD relies on the causal Markov and faithfulness assumptions and identifies the equivalence class of the underlying causal graph.

5. RAI (Yehezkel & Lerner, 2009): Recursive Autonomy Identification learns the structure by sequentially applying conditional independence tests, edge direction, and structure decomposition into autonomous sub-structures.

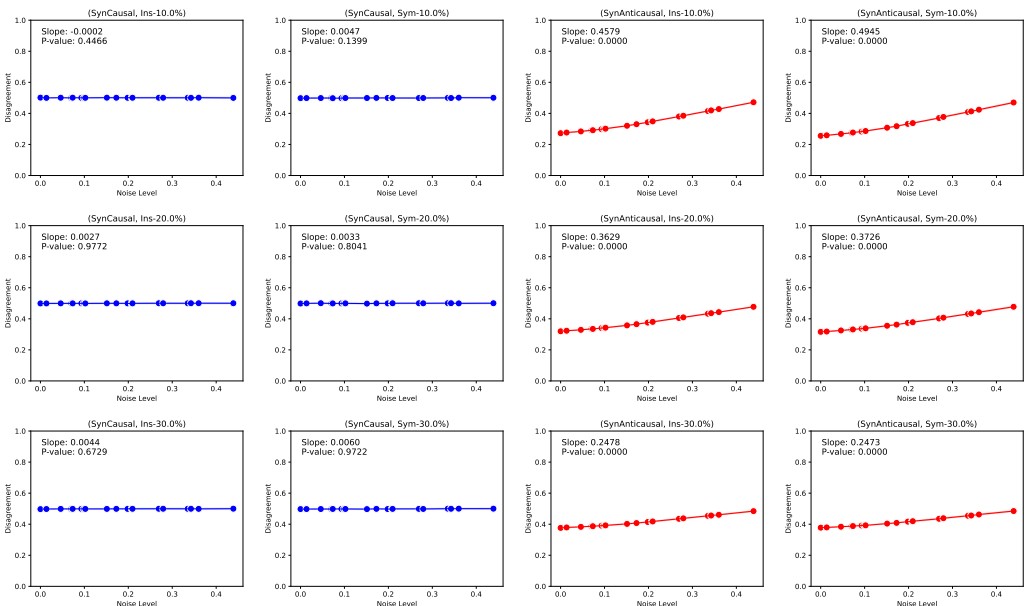

Figure 5: The average disagreement and its standard deviation under different noise rates for synthetic datasets: synCausal (left) and synAnticausal (right).

6. FCI (Spirtes et al., 2000a): Fast Causal Inference stands out among constraint-based methods for its ability to detect latent confounders.

7. LiNGAM (Shimizu et al., 2006): Linear Non-Gaussian Acyclic Model assumes that there are no hidden confounders and all of the error terms are non-gaussian and detects causal relationships from observed data accordingly.

8. SAM (Kalainathan et al., 2018): Structural Agnostic Model is a causal discovery algorithm for DAG recovery, leveraging both distributional asymmetries and conditional independencies.

9. CCDR (Aragam & Zhou, 2015): Concave Penalized Coordinate Descent with Reparametrization is a fast, score-based method for learning Bayesian networks, utilizing sparse regularization and block-cyclic coordinate descent.

## E.5 DISAGREEMENTS WITH DIFFERENT NOISE LEVELS

Fig. 5 demonstrates the trend of disagreement with $10\%$, $20\%$ and $30\%$ label errors for synCausal and synAnticausal datasets. For the *synCausal* dataset, the trend of disagreement remains unchanged at $0.5$ with the increase of noise rates, and the slope $\hat{\beta}_1$ of the regression line is close to $0$. This is because $Y'$ is poorly estimated and should be a random guess of noised $\tilde{Y}'$, which is proved in Theorem 1. On the other hand, for the *synAnticausal* dataset, there is a strong positive correlation between the disagreement and the noise level. In this case, $Y'$ is well estimated, and both $Y'$ and $\tilde{Y}$ are close to the latent (clean) class $Y$. When the noise level $\rho$ of our injected noise is increased to $0.5$, the modified label $\tilde{Y}^\rho$ becomes more seriously corrupted and tends to deviate far away from the observed label $\tilde{Y}$. This results in a larger disagreement between $\tilde{Y}^\rho$ and $Y'$.

It is also observed that the slope becomes flattered when the label-error is larger (e.g., Ins-$30\%$ and Sym-$30\%$). Under this circumstance, a large amount of original observable labels $\tilde{Y}$ are not identical to the latent clean classs $Y$ anymore. Then $\tilde{Y}$ will be closer to a random guess of the clean class. Therefore the positive correlation between $\tilde{Y}$ and $Y'$ becomes weak. However, in these extreme settings, our estimator is still robust, because the slope of our regression line is still significantly different from $0$, and we can conclude that the dataset is anticausal.

Table 3: Comparing with other baselines on synthetic and real-world datasets.

| | Method | Original | Instance | | | Pair | | | Sym | | |
|---|---|---|---|---|---|---|---|---|---|---|---|
| | | 0% | 10% | 20% | 30% | 10% | 20% | 30% | 10% | 20% | 30% |
| SynCausal (causal) | GES | causal | causal | causal | anticausal | causal | causal | anticausal | causal | causal | anticausal |
| | GIES | causal | causal | causal | anticausal | causal | causal | anticausal | causal | causal | anticausal |
| | PC | causal | causal | anticausal | anticausal | causal | anticausal | anticausal | causal | anticausal | anticausal |
| | ICD | anticausal | anticausal | anticausal | anticausal | anticausal | anticausal | anticausal | anticausal | anticausal | anticausal |
| | RAI | causal | causal | causal | causal | causal | causal | causal | causal | causal | causal |
| | FCI | anticausal | anticausal | anticausal | anticausal | anticausal | anticausal | anticausal | anticausal | anticausal | anticausal |
| | LINGAM | unknown | unknown | unknown | unknown | unknown | unknown | unknown | unknown | unknown | unknown |
| | SAM | unknown | unknown | unknown | unknown | unknown | unknown | unknown | unknown | unknown | unknown |
| | CCDR | causal | causal | causal | causal | causal | causal | causal | causal | causal | causal |
| | Our method | $p=0.7886$ causal | $p=0.3131$ causal | $p=0.4466$ causal | $p=0.6729$ causal | $p=0.1399$ causal | $p=0.8041$ causal | $p=0.9772$ causal | $p=0.1399$ causal | $p=0.8041$ causal | $p=0.9772$ causal |
| Secstr (causal) | GES | anticausal | anticausal | causal | causal | anticausal | anticausal | anticausal | anticausal | anticausal | anticausal |
| | GIES | anticausal | anticausal | causal | causal | anticausal | anticausal | anticausal | anticausal | anticausal | anticausal |
| | PC | anticausal | anticausal | anticausal | anticausal | anticausal | anticausal | anticausal | anticausal | anticausal | anticausal |
| | ICD | anticausal | anticausal | anticausal | anticausal | anticausal | anticausal | anticausal | anticausal | anticausal | anticausal |
| | RAI | unknown | unknown | unknown | unknown | unknown | unknown | unknown | unknown | unknown | unknown |
| | FCI | anticausal | anticausal | anticausal | anticausal | anticausal | anticausal | anticausal | anticausal | anticausal | anticausal |
| | LINGAM | unknown | unknown | unknown | unknown | unknown | unknown | unknown | unknown | unknown | unknown |
| | SAM | unknown | unknown | unknown | unknown | unknown | unknown | unknown | unknown | unknown | unknown |
| | CCDR | causal | causal | causal | causal | causal | causal | causal | causal | causal | causal |
| | Our method | $p=0.0000$ anticausal | $p=0.0000$ anticausal | $p=0.0000$ anticausal | $p=0.1510$ causal | $p=0.0000$ anticausal | $p=0.0000$ anticausal | $p=0.0000$ anticausal | $p=0.0000$ anticausal | $p=0.0000$ anticausal | $p=0.0000$ anticausal |
| SynAnticausal (anticausal) | GES | anticausal | anticausal | anticausal | anticausal | anticausal | anticausal | anticausal | anticausal | anticausal | anticausal |
| | GIES | anticausal | anticausal | anticausal | anticausal | anticausal | anticausal | anticausal | anticausal | anticausal | anticausal |
| | PC | anticausal | anticausal | anticausal | anticausal | anticausal | anticausal | anticausal | anticausal | anticausal | anticausal |
| | ICD | anticausal | anticausal | anticausal | anticausal | anticausal | anticausal | anticausal | anticausal | anticausal | anticausal |
| | RAI | causal | causal | causal | causal | causal | causal | causal | causal | causal | causal |
| | FCI | anticausal | anticausal | anticausal | anticausal | anticausal | anticausal | anticausal | anticausal | anticausal | anticausal |
| | LINGAM | unknown | unknown | unknown | unknown | unknown | unknown | unknown | unknown | unknown | unknown |
| | SAM | unknown | unknown | unknown | unknown | unknown | unknown | unknown | unknown | unknown | unknown |
| | CCDR | anticausal | anticausal | anticausal | causal | anticausal | anticausal | causal | anticausal | anticausal | causal |
| | Our method | $p=0.0000$ anticausal | $p=0.0000$ anticausal | $p=0.0000$ anticausal | $p=0.0000$ anticausal | $p=0.0000$ anticausal | $p=0.0000$ anticausal | $p=0.0000$ anticausal | $p=0.0000$ anticausal | $p=0.0000$ anticausal | $p=0.0000$ anticausal |
| Breastcancer (anticausal) | GES | anticausal | unknown | anticausal | unknown | unknown | anticausal | unknown | unknown | anticausal | unknown |
| | GIES | anticausal | unknown | anticausal | unknown | unknown | anticausal | unknown | unknown | anticausal | unknown |
| | PC | anticausal | unknown | anticausal | unknown | unknown | anticausal | unknown | unknown | anticausal | unknown |
| | ICD | anticausal | anticausal | anticausal | unknown | anticausal | anticausal | unknown | anticausal | anticausal | unknown |
| | RAI | anticausal | anticausal | anticausal | anticausal | anticausal | unknown | unknown | anticausal | unknown | unknown |
| | FCI | anticausal | anticausal | anticausal | unknown | anticausal | anticausal | anticausal | anticausal | anticausal | anticausal |
| | LINGAM | unknown | unknown | unknown | unknown | unknown | unknown | unknown | unknown | unknown | unknown |
| | SAM | unknown | unknown | unknown | unknown | unknown | unknown | unknown | unknown | unknown | unknown |
| | CCDR | causal | causal | causal | anticausal | causal | causal | anticausal | causal | causal | anticausal |
| | Our method | $p=0.0000$ anticausal | $p=0.0000$ anticausal | $p=0.0000$ anticausal | $p=0.0000$ anticausal | $p=0.0000$ anticausal | $p=0.0000$ anticausal | $p=0.0000$ anticausal | $p=0.0000$ anticausal | $p=0.0000$ anticausal | $p=0.0000$ anticausal |
| Coil (anticausal) | GES | anticausal | causal | anticausal | anticausal | anticausal | causal | anticausal | anticausal | anticausal | anticausal |
| | GIES | anticausal | causal | anticausal | anticausal | anticausal | causal | anticausal | anticausal | anticausal | anticausal |
| | PC | causal | unknown | causal | unknown | unknown | anticausal | causal | causal | unknown | unknown |
| | ICD | unknown | unknown | unknown | unknown | unknown | unknown | unknown | unknown | unknown | unknown |
| | RAI | unknown | unknown | unknown | unknown | unknown | unknown | unknown | unknown | unknown | unknown |
| | FCI | anticausal | anticausal | anticausal | unknown | anticausal | anticausal | anticausal | anticausal | anticausal | anticausal |
| | LINGAM | unknown | unknown | unknown | unknown | unknown | unknown | unknown | unknown | unknown | unknown |
| | SAM | unknown | unknown | unknown | unknown | unknown | unknown | unknown | unknown | unknown | unknown |
| | CCDR | causal | causal | unknown | unknown | causal | causal | causal | causal | causal | causal |
| | Our method | $p=0.0000$ anticausal | $p=1.0000$ anticausal | $p=0.0000$ anticausal | $p=0.0000$ anticausal | $p=0.0000$ anticausal | $p=0.0000$ anticausal | $p=0.0000$ anticausal | $p=0.0000$ anticausal | $p=0.0000$ anticausal | $p=0.0000$ anticausal |

## E.6 MORE EXPERIMENTS ON REAL-WORLD DATASETS

In Table 3 and 4, we present the results of causal discovery obtained using our RoCA estimator compared to other baseline methods. Our RoCA estimator outperforms the baseline methods in accurately identifying the causal relationships.

Among the 14 of 16 datasets, our RoCA estimator correctly identified the causal relationship between $X$ and $Y$ in the majority of cases. This holds even when the datasets contained different types of label noise, such as instance-dependent, pair, and symmetric noise, with noise rates ranging from $0\%$ to $30\%$. On the other hand, the performance of the baseline methods was generally satisfactory for anticausal datasets but lacked accuracy when dealing with causal datasets. This is because a causal dataset requires no features in $X$ to cause $Y$, which presents a challenge for these baseline methods. They need to ensure that there is no edge from any vertex representing features in $X$ pointing to the vertex representing $Y$ when recovering the causal diagram. Although these baseline methods tend to perform well in general tasks, they may not be suitable for this particular task, leading to misclassification of datasets as anticausal.

Furthermore, the time complexity of some baseline methods hinders their application to datasets with a large number of features, such as image datasets or datasets with hundreds of features (e.g., *G241C*, *Coil*, etc.). Completing the algorithm within a reasonable time frame becomes challenging for these methods. In this case, we classify the results as unknown.

Table 4: Comparing with other baselines on synthetic and real-world datasets (cont.).

| | Method | Original | Instance | | | Pair | | | Sym | | |
|---|---|---|---|---|---|---|---|---|---|---|---|
| | | 0% | 10% | 20% | 30% | 10% | 20% | 30% | 10% | 20% | 30% |
| G241C (anticausal) | GES | anticausal | anticausal | anticausal | anticausal | anticausal | anticausal | anticausal | causal | causal | causal |
| | GIES | anticausal | anticausal | anticausal | anticausal | anticausal | anticausal | anticausal | causal | causal | causal |
| | PC | anticausal | anticausal | unknown | anticausal | anticausal | anticausal | anticausal | anticausal | anticausal | anticausal |
| | ICD | unknown | unknown | unknown | unknown | unknown | unknown | unknown | unknown | unknown | unknown |
| | RAI | unknown | unknown | unknown | unknown | unknown | unknown | unknown | unknown | unknown | unknown |
| | FCI | anticausal | anticausal | anticausal | anticausal | anticausal | anticausal | anticausal | anticausal | anticausal | anticausal |
| | LINGAM | unknown | unknown | unknown | unknown | unknown | unknown | unknown | unknown | unknown | unknown |
| | SAM | unknown | unknown | unknown | unknown | unknown | unknown | unknown | unknown | unknown | unknown |
| | CCDR | anticausal | anticausal | anticausal | anticausal | anticausal | anticausal | anticausal | anticausal | causal | anticausal |
| | Our method | p=0.0000 | p=0.0000 | p=0.0000 | p=0.0000 | p=0.0000 | p=0.0000 | p=0.0000 | p=0.0000 | p=0.0000 | |
| | | anticausal | anticausal | anticausal | anticausal | anticausal | anticausal | anticausal | anticausal | anticausal | anticausal |
| Iris (anticausal) | GES | anticausal | unknown | unknown | unknown | unknown | unknown | unknown | unknown | unknown | unknown |
| | GIES | anticausal | unknown | unknown | unknown | unknown | unknown | unknown | unknown | unknown | unknown |
| | PC | anticausal | unknown | anticausal | unknown | anticausal | unknown | unknown | anticausal | causal | unknown |
| | ICD | anticausal | unknown | anticausal | unknown | anticausal | unknown | unknown | anticausal | anticausal | unknown |
| | RAI | anticausal | anticausal | anticausal | anticausal | anticausal | anticausal | anticausal | anticausal | anticausal | anticausal |
| | FCI | anticausal | unknown | anticausal | unknown | anticausal | unknown | unknown | anticausal | anticausal | unknown |
| | LINGAM | unknown | unknown | unknown | unknown | unknown | unknown | unknown | unknown | unknown | unknown |
| | SAM | unknown | unknown | unknown | unknown | unknown | unknown | unknown | unknown | unknown | unknown |
| | CCDR | anticausal | anticausal | anticausal | causal | anticausal | causal | causal | anticausal | anticausal | anticausal |
| | Our method | p=0.0000 | p=0.0000 | p=0.0000 | p=0.0000 | p=0.0000 | p=0.0000 | p=0.0000 | p=0.0000 | p=0.0000 | |
| | | anticausal | anticausal | anticausal | anticausal | anticausal | anticausal | anticausal | anticausal | anticausal | anticausal |
| Mushroom | GES | anticausal | anticausal | anticausal | anticausal | anticausal | anticausal | anticausal | anticausal | anticausal | anticausal |
| | GIES | anticausal | anticausal | anticausal | anticausal | anticausal | anticausal | anticausal | anticausal | anticausal | anticausal |
| | PC | causal | anticausal | anticausal | anticausal | anticausal | anticausal | anticausal | anticausal | anticausal | anticausal |
| | ICD | unknown | unknown | unknown | unknown | unknown | unknown | unknown | unknown | unknown | unknown |
| | RAI | unknown | unknown | unknown | unknown | unknown | unknown | unknown | unknown | unknown | unknown |
| | LINGAM | unknown | unknown | unknown | unknown | unknown | unknown | unknown | unknown | unknown | unknown |
| | SAM | unknown | unknown | unknown | unknown | unknown | unknown | unknown | unknown | unknown | unknown |
| | CCDR | unknown | unknown | unknown | unknown | unknown | unknown | unknown | unknown | unknown | unknown |
| | Our method | p=0.0000 | p=0.2127 | p=0.0000 | p=0.0000 | p=0.0000 | p=0.0000 | p=0.0865 | p=0.0000 | p=0.0000 | p=0.0865 |
| | | anticausal | causal | anticausal | anticausal | anticausal | anticausal | causal | anticausal | anticausal | causal |
| Segment (anticausal) | GES | unknown | anticausal | anticausal | anticausal | anticausal | anticausal | anticausal | anticausal | anticausal | anticausal |
| | GIES | unknown | anticausal | anticausal | anticausal | anticausal | anticausal | anticausal | anticausal | anticausal | anticausal |
| | PC | unknown | anticausal | anticausal | anticausal | anticausal | anticausal | anticausal | anticausal | anticausal | anticausal |
| | ICD | anticausal | anticausal | anticausal | anticausal | anticausal | anticausal | anticausal | anticausal | anticausal | anticausal |
| | RAI | unknown | unknown | unknown | unknown | unknown | unknown | unknown | unknown | unknown | unknown |
| | FCI | anticausal | anticausal | anticausal | anticausal | anticausal | anticausal | anticausal | anticausal | anticausal | anticausal |
| | LINGAM | unknown | unknown | unknown | unknown | unknown | unknown | unknown | unknown | unknown | unknown |
| | SAM | unknown | unknown | unknown | unknown | unknown | unknown | unknown | unknown | unknown | unknown |
| | CCDR | causal | causal | anticausal | anticausal | anticausal | anticausal | anticausal | anticausal | anticausal | anticausal |
| | Our method | p=0.0000 | p=0.0000 | p=0.0000 | p=0.0000 | p=0.0000 | p=0.0000 | p=0.0000 | p=0.0000 | p=0.0000 | |
| | | anticausal | anticausal | anticausal | anticausal | anticausal | anticausal | anticausal | anticausal | anticausal | anticausal |
| Usps (anticausal) | GES | anticausal | anticausal | causal | anticausal | causal | anticausal | causal | causal | anticausal | causal |
| | GIES | anticausal | anticausal | causal | anticausal | causal | anticausal | causal | causal | anticausal | causal |
| | PC | causal | unknown | unknown | unknown | causal | anticausal | unknown | causal | anticausal | unknown |
| | ICD | unknown | unknown | unknown | unknown | unknown | unknown | unknown | unknown | unknown | unknown |
| | RAI | unknown | unknown | unknown | unknown | unknown | unknown | unknown | unknown | unknown | unknown |
| | FCI | anticausal | anticausal | anticausal | unknown | anticausal | anticausal | anticausal | anticausal | anticausal | anticausal |
| | LINGAM | unknown | unknown | unknown | unknown | unknown | unknown | unknown | unknown | unknown | unknown |
| | SAM | unknown | unknown | unknown | unknown | unknown | unknown | unknown | unknown | unknown | unknown |
| | CCDR | causal | causal | causal | causal | causal | causal | causal | causal | causal | causal |
| | Our method | p=0.0000 | p=0.0000 | p=0.0000 | p=0.0000 | p=0.0000 | p=0.0000 | p=0.0000 | p=0.0000 | p=0.0000 | |
| | | anticausal | anticausal | anticausal | anticausal | anticausal | anticausal | anticausal | anticausal | anticausal | anticausal |
| Waveform (anticausal) | GES | anticausal | causal | causal | causal | causal | causal | causal | causal | anticausal | causal |
| | GIES | anticausal | causal | causal | causal | causal | causal | causal | causal | anticausal | causal |
| | PC | anticausal | anticausal | anticausal | anticausal | anticausal | anticausal | anticausal | anticausal | anticausal | anticausal |
| | ICD | anticausal | anticausal | anticausal | anticausal | anticausal | anticausal | anticausal | anticausal | anticausal | anticausal |
| | RAI | anticausal | unknown | unknown | unknown | unknown | unknown | unknown | unknown | unknown | unknown |
| | FCI | anticausal | anticausal | anticausal | anticausal | anticausal | anticausal | anticausal | anticausal | anticausal | anticausal |
| | LINGAM | unknown | unknown | unknown | unknown | unknown | unknown | unknown | unknown | unknown | unknown |
| | SAM | unknown | unknown | unknown | unknown | unknown | unknown | unknown | unknown | unknown | unknown |
| | CCDR | causal | causal | causal | causal | causal | anticausal | causal | causal | anticausal | causal |
| | Our method | p=0.0000 | p=0.0000 | p=0.0000 | p=0.0000 | p=0.0000 | p=0.0000 | p=0.0000 | p=0.0000 | p=0.0000 | |
| | | anticausal | anticausal | anticausal | anticausal | anticausal | anticausal | anticausal | anticausal | anticausal | anticausal |
| Digit1 (anticausal) | GES | anticausal | causal | causal | anticausal | causal | causal | causal | causal | causal | causal |
| | GIES | anticausal | causal | causal | anticausal | causal | causal | causal | causal | causal | causal |
| | PC | causal | anticausal | anticausal | causal | anticausal | unknown | unknown | anticausal | unknown | unknown |
| | ICD | unknown | unknown | unknown | unknown | unknown | unknown | unknown | unknown | unknown | unknown |
| | RAI | unknown | unknown | unknown | unknown | unknown | unknown | unknown | unknown | unknown | unknown |
| | FCI | anticausal | anticausal | anticausal | anticausal | anticausal | anticausal | anticausal | anticausal | anticausal | anticausal |
| | LINGAM | unknown | unknown | unknown | unknown | unknown | unknown | unknown | unknown | unknown | unknown |
| | SAM | unknown | unknown | unknown | unknown | unknown | unknown | unknown | unknown | unknown | unknown |
| | CCDR | anticausal | anticausal | anticausal | anticausal | anticausal | anticausal | anticausal | anticausal | anticausal | anticausal |
| | Our method | p=0.0000 | p=0.0000 | p=0.0000 | p=0.0000 | p=0.0000 | p=0.0000 | p=0.0000 | p=0.0000 | p=0.0000 | |
| | | anticausal | anticausal | anticausal | anticausal | anticausal | anticausal | anticausal | anticausal | anticausal | anticausal |

## F    PSEDUOCODE OF OUR INSTANCE-DEPENDENT NOISE GENERATION

---

**Algorithm 1** Generation of Instance-dependent Noisy Labels

---

**Require:**  An average noise level $\rho$; A sample $S = \{(X_i, \tilde{Y}_i)\}_{i=0}^{m}$, where contains $C$ number of classes, and $X \in \mathbb{R}^d$.

1: Initialize an empty list $A$ with length $m$.
2: **for** each $i^{th}$ example $(x_i, \tilde{y}) \in S$: **do**
3:     Let $a_i = ||X_i||_1$ and add $a_i$ into $A$.
4: **end for**
5: Sort the values in $A$ in ascending order.
6: Sample a vector $P \in \mathbb{R}^m$ from a $m$-dimensional truncated normal distribution with mean $\rho$, upper limit 1, and lower limit 0.
7: Sort the values in $P$ in ascending order.
8: **for** $i$ in range $(0, m)$: **do**
9:     Let the individual flip rate of the $i^{th}$ example $\rho_{x_i}$= (the $i^{th}$ element in $P$).
10: **end for**
11: Generate the instance-dependent noisy label of the $i^{th}$ example $\tilde{Y}^{\rho_{x_i}}$ using the flip rate $\rho_{x_i}$.

---

## G    PROOFS

In this section, we show all the proofs. We remind some notations first.

- Let $\mathcal{X}$ be the instance space and $C$ the set of all possible classes.
- Let $S = \{(x_i, \tilde{y}_i)\}_{t=0}^{m}$ be an sample set.
- Let $h : \mathcal{X} \to \{1, \ldots, C\}$, be a hypothesis that predicts pseudo labels of instances. Concretely, it can be a K-means algorithm together with the Hungarian algorithm which matches the cluster ID to the corresponding pseudo labels. Let $\mathcal{H}$ be the hypothesis space, where $h \in \mathcal{H}$.
- Let $\tilde{R}^\rho(h) = \mathbb{E}_{(\boldsymbol{x}, \tilde{y}^\rho) \sim P(\boldsymbol{X}, \tilde{Y}^\rho)}[\mathbb{1}_{\{h(\boldsymbol{x}) \neq \tilde{y}^\rho\}}]$be the expected disagreement $\tilde{R}(h)$ between pseudo labels and generated labels $\tilde{y}^\rho$ with $\rho$-level noise injection.
- Let $\hat{\tilde{R}}_S^\rho(h)$ be the average disagreement (or empirical risk) of $h$ on the set $S$ after $\rho$-level noise injection.

Firstly, we illustrate the Rademacher complexity bound.

**Definition 3** (The Rademacher Complexity Bound (Mohri et al., 2018))**.** Let $\mathcal{H}$ be a family of functions taking values in $\{-1, +1\}$, and let $\mathcal{D}$ be the distribution over the input space $\mathcal{X}$. Then, for any $\delta > 0$, with probability at least $1 - \delta/2$ over a sample $S = (x_1, \ldots, x_m)$ of size $m$ drawn according to $\mathcal{D}$, for any function $h \in \mathcal{H}$,

$$\hat{R}_S(h) - R(h) \leq 2\hat{\mathfrak{R}}_S(\mathcal{H}) + 3\sqrt{\frac{\log \frac{4}{\delta}}{2n}}, \tag{8}$$

where $R(h)$ is the expected risk of the function $h$, and $\hat{R}_S(h)$ is the empirical risk of the function $h$ on the sample $S$ (Mohri et al., 2018). Specifically, let $c$ be a target concept, then,

$$R(h) = \mathbb{E}_{x \sim \mathcal{D}}[\mathbb{1}_{\{h(x_i) \neq c(x_i)\}}], \ \hat{R}_S(h) = \frac{1}{m} \sum_{i=1}^{m} \mathbb{1}_{\{h(x_i) \neq c(x_i)\}}.$$

### G.1    PROOF OF THEOREM 1

*Proof.* Under causal setting, $h$ random guess the clean labels, i.e., $\forall i, j \in C \wedge i \neq j \wedge \forall t \in \{0, 1, \ldots, m\}$, $P(Y' = y'|Y = y, X = x) = \frac{1}{C}$. Then we will prove that if $h$ can only random guess the clean labels, then $h$ can only random guess the observed labels that contain the label error, i.e., $\tilde{R}(h, x) = \frac{C-1}{C}$.

By the assumption that for every instance and clean class pair $(x, y)$, its observed label $\tilde{y}$ is obtained by a noise rate $\rho_x$ such that $P(\tilde{Y} = \tilde{y}|Y = y, \boldsymbol{X} = x) = \frac{\rho_x}{C-1}$ for all $\tilde{y} \neq y \wedge \tilde{y} \in C$, the risk $\tilde{R}(h, x)$ of $h$ on $x$ and its observed labels comes from two parts:

- When $h$ misclassifies the clean label, $h$ also misclassifies the observed label, i.e., ($y \neq y'$ and $y' \neq \tilde{y}$).

- When $h$ successfully classifies the clean label, $h$ misclassifies the observed label, i.e., ($y = y'$ and $y' \neq \tilde{y}$).

Specifically, the expected risk of each example is as follows.

$$
\begin{aligned}
\tilde{R}(h, x) &= R(h, x)(1 - \frac{\rho_x}{C - 1}) + (1 - R(h, x))\rho_x \\
&= \frac{C - 1}{C} \frac{C - 1 - \rho_x}{C - 1} + \frac{\rho_x}{C} \\
&= \frac{C - 1 - \rho_x}{C} + \frac{\rho_x}{C} \\
&= \frac{C - 1}{C}.
\end{aligned} \tag{9}
$$

Because our noise is designed to also satisfy the assumption, after injecting our designed instance-dependent noise, the risk $\hat{\tilde{R}}_S^{\rho^1}(h)$ and $\hat{\tilde{R}}_S^{\rho^2}(h)$ under two different (expected) noise levels $\rho^1 = \mathbb{E}_x[\rho_x^1]$ and $\rho^2 = \mathbb{E}_x[\rho_x^2]$ does not change under the anticausal setting.

$$
\tilde{R}^{\rho^1}(h, x) = \tilde{R}(h, x)(1 - \frac{\rho_x^1}{C - 1}) + (1 - \tilde{R}(h, x))\rho_x^1 = \frac{C - 1}{C}. \tag{10}
$$

$$
\tilde{R}^{\rho^2}(h, x) = \tilde{R}(h, x)(1 - \frac{\rho_x^2}{C - 1}) + (1 - \tilde{R}(h, x))\rho_x^2 = \frac{C - 1}{C}. \tag{11}
$$

The above equations show that after injecting two different levels of instance-dependent label noise, the risks do not change. For completeness, we also illustrate the convergence rate of the difference between two empirical risks with respect to sample size. By employing the Rademacher complexity bound, with a probability $1 - \delta/2$,

$$
\begin{aligned}
\hat{\tilde{R}}_S^{\rho^1}(h) &\leq \mathbb{E}_x[\tilde{R}^{\rho^1}(h, x)] + 2\hat{\mathfrak{R}}_S(\mathcal{H}) + 3\sqrt{\frac{\log\frac{4}{\delta}}{2m}} \\
&= \tilde{R}^{\rho^1}(h) + 2\hat{\mathfrak{R}}_S(\mathcal{H}) + 3\sqrt{\frac{\log\frac{4}{\delta}}{2m}},
\end{aligned}
$$

similarly,

$$
\begin{aligned}
\hat{\tilde{R}}_S^{\rho^2}(h) &\leq \mathbb{E}_x[\tilde{R}^{\rho^2}(h, x)] + 2\hat{\mathfrak{R}}_S(\mathcal{H}) + 3\sqrt{\frac{\log\frac{4}{\delta}}{2m}} \\
&= \tilde{R}(h, \rho^2) + 2\hat{\mathfrak{R}}_S(\mathcal{H}) + 3\sqrt{\frac{\log\frac{4}{\delta}}{2m}}.
\end{aligned}
$$

By applying the symmetric property of the Rademacher complexity bound to the above two inequalities, with a probability $1 - 2\delta$,

$$
|\hat{\tilde{R}}_S^{\rho^1}(h) - \tilde{R}^{\rho^1}(h)| \leq 2\hat{\mathfrak{R}}_S(\mathcal{H}) + 3\sqrt{\frac{\log\frac{4}{\delta}}{2m}} \text{ and}
$$

$$
|\hat{\tilde{R}}_S^{\rho^2}(h) - \tilde{R}(h, \rho^2)| \leq 2\hat{\mathfrak{R}}_S(\mathcal{H}) + 3\sqrt{\frac{\log\frac{4}{\delta}}{2m}}.
$$

Combining the above two inequalities, we get

$$|\hat{\tilde{R}}_S^{\rho^1}(h) - \tilde{R}^{\rho^1}(h) - \hat{\tilde{R}}_S^{\rho^2}(h) + \tilde{R}(h, \rho^2)| \le 4\hat{\mathfrak{R}}_S(\mathcal{H}) + 6\sqrt{\frac{\log \frac{4}{\delta}}{2m}}.$$

By Eq. 10 and Eq. 11, the expected risk $\tilde{R}^{\rho^1}(h) = \mathbb{E}_X\left[\tilde{R}^{\rho^1}(h, x)\right] = \mathbb{E}_X\left[\frac{C-1}{C}\right]$ and $\tilde{R}(h, \rho^2) = \mathbb{E}_X\left[\tilde{R}^{\rho^2}(h, x)\right] = \mathbb{E}_X\left[\frac{C-1}{C}\right]$ both equals to $\frac{C-1}{C}$, then the above inequality becomes

$$|\hat{\tilde{R}}_S^{\rho^1}(h) - \hat{\tilde{R}}_S^{\rho^2}(h)| \le 4\hat{\mathfrak{R}}_S(\mathcal{H}) + 6\sqrt{\frac{\log \frac{4}{\delta}}{2m}}, \tag{12}$$

with a probability 1-2$\delta$, which completes the proof. □

### G.2 PROOF OF THEOREM 2

*Proof.* The expected risk on the observed label for each instance $x$ is that:

$$\tilde{R}^{\rho}(h, x) = \tilde{R}(h, x)(\frac{\rho_x}{C-1}) + (1 - \tilde{R}(h, x))(1 - \rho_x)$$

$$= \rho_x + \tilde{R}(h, x) - \rho_x \tilde{R}(h, x) - \frac{\rho_x \tilde{R}(h, x)}{C-1}.$$

Then the expected risk of the distribution of observed data is that:

$$\tilde{R}^{\rho}(h) = \mathbb{E}_X\left[\tilde{R}^{\rho}(h, x)\right]$$

$$= \mathbb{E}_X\left[\rho_x + \tilde{R}(h, x) - \rho_x \tilde{R}(h, x) - \frac{\rho_x \tilde{R}(h, x)}{C-1}\right]$$

$$= \mathbb{E}_X\left[\tilde{R}(h, x)\right] + \mathbb{E}_X\left[\rho_x - \rho_x \tilde{R}(h, x) - \frac{\rho_x \tilde{R}(h, x)}{C-1}\right]$$

$$= \tilde{R}(h) + \mathbb{E}_X\left[\left(1 - \tilde{R}(h, x) - \frac{\tilde{R}(h, x)}{C-1}\right)\rho_x\right]$$

$$= \tilde{R}(h) + \mathbb{E}_X\left[\left(1 - \frac{C\tilde{R}(h, x)}{C-1}\right)\rho_x\right]. \tag{13}$$

Moving $\tilde{R}(h)$ to the LHS of the above equation completes the proof. □

Note that the convergence rate of $\mathbb{E}\left[\left(1 - \frac{C\tilde{R}(h,x)}{C-1}\right)\rho_x\right]$ can also be directly derived by replacing the expected risks with the empirical risks in Eq. 13. This process employs Inequality 8 and shares a similar conceptual foundation as the proof for the coverage rate presented in Theorem 1.

## H    DISCUSSION & LIMITATION

It is known that discovering causal relations without any assumptions is impossible (Glymour et al., 2019a). Here, we discuss the assumptions required by our method. Except for the commonly used causal faithfulness and acyclic graph assumption and independent causal mechanisms (Peters et al., 2014) to guarantee that by employing RoCA under the causal setting, the disagreements (or expected risks) under different noise levels do not change, an additional assumption is required to constrain the label error types in datasets. Specifically, this assumption states that for every instance and clean class pair $(x, y)$, its observed label $\tilde{y}$ is obtained by a noise rate $\rho_x$ such that $P(\tilde{Y} = \tilde{y}|Y = y, \boldsymbol{X} = x) = \frac{\rho_x}{C-1}$ for all $\tilde{y} \neq y \wedge \tilde{y} \in C$. This assumption holds not only when

data contains instance-dependent label errors but also when data is without label errors or contains class-dependent errors (Patrini et al., 2017; Xia et al., 2019; Li et al., 2021).

To let our method successfully infer an anticausal dataset, another assumption is that the anticausal dataset should contain information about the variables that are effects of the clean class $Y$. If the information does not exist, then the distribution of instances $P(\boldsymbol{X})$ will no longer contain information about $P(Y|\boldsymbol{X})$. In this case, RoCA will misclassify it as a causal dataset. It is important to note that the effect feature itself can be latent, but its information has to be retained in the dataset. For example, consider an image dataset, even though content and style features are latent, they are inherently incorporated into the image itself (Von Kügelgen et al., 2021), ensuring that information on content and style features is retained in the dataset.

Furthermore, to use $P_\theta(\tilde{Y}|\boldsymbol{X})$ as a surrogate for $P(Y|\boldsymbol{X})$, we assume that there is a dependence between $P_\theta(\tilde{Y}|\boldsymbol{X})$ and $P_\theta(Y|\boldsymbol{X})$. Note that this assumption usually holds, as otherwise the annotation mechanism $P_\theta(\tilde{Y}|\boldsymbol{X})$ would just be a random guess of $P(Y|\boldsymbol{X})$, resulting in the observed label $\tilde{Y}$ becoming meaningless. However, if the assumption does not hold, employing $P_\theta(\tilde{Y}|\boldsymbol{X})$ as the surrogate will lead to misclassification of causal and anticausal relation.

