# OpenReview forum: "RoCA: A Robust Method to Discover Causal or Anticausal Relation by Noise Injection"
_ICLR.cc/2024/Conference — Submitted to ICLR 2024_

### Official Review · Reviewer_aduf · 2023-10-29

**Soundness:** 3 good
**Presentation:** 3 good
**Contribution:** 2 fair
**Rating:** 6
**Confidence:** 2

**Summary:**

This paper considers a bivariate causal discovery problem where the observed labels are noisy. Observing that in the causal direction $P(X)$ does not contain information about the mechanism $P(Y|X)$, authors show that $P(\tilde Y|X)$ is a good surrogate in this setting and then propose a noise injection based method to discover the causal direction. Some theoretical results are given and experimental results validate the proposed method.

**Strengths:**

- The method of noise injection for causal discovery is novel and interesting.
- Theoretical guarantees are given.
- Illustration of the proposed method is good.

**Weaknesses:**

- Problem setting is not very rigorous
- No explicit identification result and no assumptions/conditions under which the proposed method can identifiy the true direction.
- Lacking some details regarding the experiment part.

**Questions:**

I reviewed this work in a previous venue where authors have addressed many of my concerns. For this version, I only have the following questions/suggestions:

- Compared with traditional biavariate causal discovery methods, this work and the proposed method are novel in two aspects: 1) noisy labels, and 2) high-dim features $X$ and scalar label $Y$. Thus, I think the paper should be made more clear regarding the problem setting and assumptions in Section 3.1:
  - "This mechanism, being correlated with P(Y|X), provides insights into the true class posterior."---how do you define "correlated"? and in what sense?
  - "It’s noteworthy that Pθ(Y˜ |X)  generally maintain a dependence with Pθ(Y |X)." Similarly, how do you define "dependence"  in a more accurate way (e.g., using math formulas) and in what sense?
  - "It is usually highly correlated with and informative about P(Y |X). Moreover, under a causal setting, P(X) cannot inform P(Y˜ |X), since Y˜ and Y are effects of X, and P(X) and P(Y˜ |X) follows causal factorization and are disentangled according to independent mechanisms (Peters et al., 2017b). Thus, Pθ(Y˜ |X) is an proper surrogate." -----From the causal graph in Fig. 2, I can see an edge $X->\tilde Y$  for both causal and anticausal settings, so the factorization should work for both directions. Or three should be a more accurate causal graph regarding the proposed setting?

- There seems no identification conditions  discussed in the main text? Please make identification explicit in the main text and discuss also the assumptions. This would help readers have a better understanding of the proposed method.
- Experiments: please do consider to have more details in the main text (e.g., in the camera-ready version where more pages are allowed or move some content to the appendix) For example, Table 1 seems not mentioned at all in the main text.
- A minor question: GES and some other methods work for scalar variables. How do you apply these methods to image data? I do no find such details and cannot say if the comparison with these methods are proper.

---

> ### Author Response · Authors · 2023-11-16
> **Rebuttal by Authors (Part 1)**
>
> **Q1. Define dependence  between $P_\theta(\tilde{Y} |\boldsymbol{X})$ and $P(Y |\boldsymbol{X})$**
>
>
>
> **A1.** To define the dependence between $P_\theta(\tilde{Y} |\boldsymbol{X})$ and $P(Y |\boldsymbol{X})$, we first consider the nature of the observed label $\tilde{Y}$ in relation to the underlying clean class $Y$. Intuitively, if $\tilde{Y}$ is meaningful, it should be an estimation of $Y$ and not just a completely random guess. This relationship can be expressed mathematically in terms of conditional probabilities.
>
> - If $P_\theta(\tilde{Y} |\boldsymbol{X})$ and $P_\theta(Y |\boldsymbol{X})$ are completely independent, then for every instance $\boldsymbol{X}$, the probability $P(\tilde{Y}=\tilde{y} |Y =y , \boldsymbol{X}=\boldsymbol{x})$ equals $1/C$, where $C$ is the number of classes. This implies that for each instance $\boldsymbol{X}$, its observed label $\tilde{Y}$ is essentially a random guess of its clean class $Y$.
>
> - However, if $P_\theta(\tilde{Y} |\boldsymbol{X})$ and $P_\theta(Y |\boldsymbol{X})$ are dependent, then typically, $P(\tilde{Y}=\tilde{y} |Y =y , \boldsymbol{X}=\boldsymbol{x}) \neq 1/C$. In such cases, the observed label $\tilde{Y}$ provides information about the clean class $Y$, indicating a non-random relationship between the observed and true labels.
>
>
> **Q2. What is the meaning of $P_\theta(\tilde{Y} |\boldsymbol{X})$ and $P(Y |\boldsymbol{X})$ are correlated.**
>
>
> **A2.** Thank you very much for pointing this out. It means $P_\theta(\tilde{Y} |\boldsymbol{X})$ and $P_\theta(Y |\boldsymbol{X})$ are dependent. We have changed all the words “correlated” to “dependent” to improve the consistency.
>
>
>
> **Q3.  From the causal graph in Fig. 2, I can see an edge for both causal and anticausal settings, so the factorization should work for both directions.**
>
> **A3.** We apologize for any confusion. It is important to note that $\tilde{Y}$ is a different variable from $Y$. It represents the random variable for the observed label.
>
> The primary purpose of Fig. 2 is to illustrate that $P_{\theta}(\tilde{Y}|\boldsymbol{X})$ usually contains information about $P(Y|\boldsymbol{X})$. It is not a causal graph but is used to demonstrate that annotators typically learn $P_{\theta}(\tilde{Y}|\boldsymbol{X})$ using a small set of clean labeled examples. Consequently, $P_{\theta}(\tilde{Y}|\boldsymbol{X})$ should contain some information about $P(Y|\boldsymbol{X})$.
>
> To prevent confusion,
>
> - we have updated the caption of Figure 2 from "An illustration of the data generative process contains label errors" to "An illustration of annotation involving label errors", to not emphasize the data generative process, which may lead readers to think it is a causal graph.

---

> ### Author Response · Authors · 2023-11-16
> **Rebuttal by Authors (Part 2)**
>
> **Q4. Discuss the assumptions identifiable in the main text.**
>
> **A4.** Thank you for pointing this out. To accommodate the changes, we have moved the paragraph "Causal Graphs and Structural Causal Models (SCM)" to the Appendix. We have also adjusted the "Discussion & Limitation" section, now placing it in the main text at the end of the "Theoretical Analyses" section. The specific adjustments are as follows:
> ________________________
> > **Assumptions for Discovering Causal and Anticausal Relationships**
> Our method is based on commonly accepted assumptions in causal discovery: causal faithfulness, acyclic graph assumption, absence of latent confounders, and independent causal mechanisms  (Peters et al., 2014).
> To ensure that the disagreements (or expected risks) under different noise levels remain constant in a causal setting when employing RoCA, an additional assumption is needed to constrain the types of label errors in datasets.
> Specifically, this assumption states that for every instance and clean class pair $(x, y)$, the observed label $\tilde{y}$ is derived with a noise rate $\rho_x$ such that $P(\tilde{Y}=\tilde{y}|Y=y,\boldsymbol{X}=x)=\frac{\rho_{x}}{C-1}$ for all $\tilde{y} \neq y \land \tilde{y} \in C$. This assumption is applicable not only when data contains instance-dependent label errors but also when there are no label errors or when data contains class-dependent errors \cite{patrini2017making, xia2019anchor, li2021provably}.
>
> > Furthermore, to use $P_{\theta}(\tilde{Y}|\boldsymbol{X})$ as a surrogate for $P(Y|\boldsymbol{X})$, we assume a dependence between $P_{\theta}(\tilde{Y}|\boldsymbol{X})$ and $P(Y|\boldsymbol{X})$. This assumption usually holds as the absence of such a dependence would imply that the annotation mechanism $P_{\theta}(\tilde{Y}|\boldsymbol{X})$ is just a random guess of $P(Y|\boldsymbol{X})$, making the observed label $\tilde{Y}$ meaningless.
>
> > Additionally, the effectiveness of our method can be influenced by the choice of a backbone clustering method. Specifically, when handling an anticausal dataset, our approach depends on a clustering method that is capable of extracting relevant information from $P(\boldsymbol{X})$ to predict $P(Y|\boldsymbol{X})$, rather than simply guessing randomly. Owing to recent advances in unsupervised and self-supervised methods, some approaches (Niu et al., 2021) based on contrastive learning have achieved performance competitive with supervised methods on benchmark image datasets like CIFAR10 and MNIST.
> ________________________
>
>
>
> **Q5. GES and some other methods work for scalar variables. How do you apply these methods to image data?**
>
> **A5.** These methods cannot be applied to image datasets. This limitation presents an advantage of our method, which can be applied to large-scale datasets containing high-dimensional features. For instance, Clothing1M is a dataset that comprises 1 million images, and our method is suited to handle such data.

---

> ### Author Response · Authors · 2023-11-16
> **Rebuttal by Authors (Part 3)**
>
> **Q6. please do consider having more details of experiments in the main text.**
>
> **A6.** Thank you very much for your suggestion. We apologize for any inconvenience caused by the compression of the experiments section due to space limitations. We have made our best effort to include more details:
>
> 1. To clarify how the p-value is obtained in Tables, we have shifted and revised the discussion from the Method section to the Experiments section. This allows readers to quickly grasp the main idea behind the p-value calculation. The revised section is as follows:
> __________________
> > To rigorously validate the disagreement, rather than directly evaluating if the slope $\hat{\beta}_1$ contained from Eq. (4) is near $0$, we use a hypothesis test on the slope.
> The level of noise $\rho$ is randomly sampled $20$ times from a range between $0$ and $0.5$. For every selected noise level, a disagreement between $Y'$ and $\tilde{Y}$ can be calculated. Consequently, a slope value is calculated from the correlation of these $20$ noise levels and their respective disagreement ratios. By repeating such a procedure for $30$ times, a set of slope values can be obtained. These slope values are then utilized in our hypothesis test to verify if the average slope is significantly different from $0$. The details of this test are provided in Appendix A. Intuitively, if the resulting $p$-value from the test exceeds $0.05$, the slope is likely to close to zero, indicating a causal dataset. Conversely, a $p$-value below $0.05$ suggests an anticausal dataset.
>
> __________________
>
>
> 2. We have revised paragraph Disagreements with Different Noise Levels on Synthetic Datasets and Disagreements with Different Noise Levels on Synthetic Datasets to contain as much information under limited space as follows.
> __________________
> >**Disagreements with Different Noise Levels on Synthetic Datasets**
> Fig. 4 demonstrates the trend of disagreement with different noise levels for *synCausal* and *synAnticausal* datasets. To construct datasets with label errors,  $30\%$ label errors are added into these datasets. For the *synCausal* dataset, the trend of disagreement remains unchanged at $0.5$ with the increase of injected noise rates, and the slope $\hat{\beta}_1$ of the regression line is close to $0$. This is because $Y'$ is poorly estimated and should be a random guess of noised $\tilde{Y}"$.
>
> > On the other hand, for the *synAnticausal* dataset with small label errors (e.g., Sym and Ins-$10\%$ to $20\%$), there is a strong positive correlation between the disagreement and the noise level. In this case, $Y'$ is better than a random guess of both $\tilde{Y}$ and the latent clean class $Y$. Specifically, with the increase of noise level $\rho$, the corresponding $\tilde{Y}^{\rho}$ becomes more seriously polluted and tends to deviate far away from the observed label $\tilde{Y}$. This results in a larger disagreement between $\tilde{Y}^{\rho}$ and $Y'$.
>
> >**Performance of RoCA on Real-World Datasets**
> \textcolor{blue}{We have also benchmarked the RoCA method against other causal discovery algorithms. Our results, as presented in Tab. 1 and Tab. 2, demonstrate that our method is both more accurate and robust. In these tables, the term "unknown" indicates cases where the algorithm either failed to detect the causal relation, or did not complete the analysis within a feasible running time. Note that only RoCA can applied to image datasets CIFAR10N and Clothing1M to detect causal and anticausal relations.}
> ________________

---

> > ### Comment · Reviewer_aduf · 2023-11-18
> >
> > Thanks for response. Two questions remain:
> >
> > - Regarding Q3: my question was about the edge $X\to \hat Y$ in Fig. 2. I did not see variables $X_2$ and $X_4$ in Fig. 2. Please clarify.
> >
> > - Regarding GES: so GES was not applied in the image-data experiments, right?

---

> ### Author Response · Authors · 2023-11-18
> **Response by Authors to Follow-Up Questions**
>
> Dear Reviewer aduf,
>
> Thank you very much for the timely feedback. Here are the responses to the follow-up questions.
>
> **Follow-up Q1: In Fig. 2, there is an edge $\boldsymbol{X}\to \tilde{Y}$ for both causal and anticausal settings.**
>
> Follow-up A1: We apologize for any confusion. It is important to note that $\tilde{Y}$ is a different variable from $Y$. It represents the random variable for the observed label.
>
> The purpose of Fig. 2 is to illustrate that $P_{\theta}(\tilde{Y}|\boldsymbol{X})$ usually contains information about $P(Y|\boldsymbol{X})$. It is not a causal graph but is used to demonstrate that annotators typically learn $P_{\theta}(\tilde{Y}|\boldsymbol{X})$ using a small set of clean labeled examples. Consequently, $P_{\theta}(\tilde{Y}|\boldsymbol{X})$ should contain some information about $P(Y|\boldsymbol{X})$.
>
> To prevent confusion,
>
> - we have updated the caption of Figure 2 from *"An illustration of the data generative process contains label errors"* to *"An illustration of annotation involving label errors"*, to not emphasize the data generative process, which may lead readers to think it is a causal graph.
>
> - *We have also updated our original answer in the rebuttal*.
>
> Please kindly let us know if there are any other concerns. Thank you very much for pointing this out.
>
>
>
> **Follow-up Q2: Is GES  applied in the image-data experiments?**
>
> Follow-up A2: No, GES is not used in the image dataset experiments.
>
> Specifically, when attempting to apply GES to image data, GES would need to treat $\boldsymbol{X}$ as a single high-dimensional variable. Consequently, this reduces the scenario to just two variables: $\boldsymbol{X}$ and $Y$. However, GES requires at least three variables to detect causal direction.
>
>
> Kind regards,
>
> Authors

---

### Official Review · Reviewer_g7sE · 2023-10-31

**Soundness:** 2 fair
**Presentation:** 2 fair
**Contribution:** 2 fair
**Rating:** 6
**Confidence:** 4

**Summary:**

This paper introduces RoCA, a method of determining whether the causal direction between features $X$ and label $Y$ is causal or anticausal. The observed labels $\tilde{Y}$ are a noisy proxy for the true labels $Y$. The approach assigns a pseudo-label $Y’$ to each datapoint based on a clustering of the feature space and then decides that the dataset is causal if $Y’$ cannot be predicted from the features $X$ and observed label $\tilde{Y}$ (implying that $Y’$ contains no information about $P(\tilde{Y} \mid X)$). This decision is done tractably by selectively adding noise to the observed labels $\tilde{Y}$ depending on the features $X$, then observing the disagreement between the pseudo-labels $Y’$ and the noisy labels at different levels of noise. The argument for this approach is that if $Y’$ is not informative of $P(\tilde{Y} \mid X)$, then adding noise would not change anything. However, if it is informative, then adding noise would make $\tilde{Y}$ increasingly unpredictable. Experimental results show that the approach is more accurate than competing approaches.

**Strengths:**

1. The theoretical analysis on the noise injection levels is quite interesting and insightful.

2. The hypothesis testing makes the end decision more quantifiable and systematic.

3. The experimental results are quite impressive.

**Weaknesses:**

1. The way that the concept of independent mechanisms is presented in this paper seems misleading.
First, the paper quotes at the bottom of page 3 that “the mechanism generating the effect from its cause does not contain any information about the mechanism generating the cause” and then concludes that “the conditional distributions of each variable, given all causal parents, are independent entities that do not share any information”. This conclusion only holds under the Markovianity assumption (i.e. no unobserved confounders). Under the presence of unobserved confounding, it is possible that the causal mechanisms can be independent, but a variable can still be dependent on some other variable given its parents. It seems that this assumption is actually key to the effectiveness of the proposed approach. However, this assumption is not stated anywhere in the paper.
Second, mathematically speaking, $P(X, Y)$ can be factorized as either Eq. 2 or Eq. 3 regardless of causal orientation. It is not clear what is the causal consequence of choosing one over the other.
Third, it is not formally explained what it means for a distribution to “inform” another, yet this seems to be key to understanding the proposed approach. Is the paper claiming to somehow infer the data-generating mechanisms from the distributions?

2. I am concerned about the soundness of the approach. The approach seems to rely on the property that within causal datasets, observed labels are evenly distributed among the clusters of $P(X)$, while they are not in anticausal datasets. This property does not seem to be related to any causal properties.

3. Assumptions are quite unclear. In addition to the assumption of Markovianity, it seems there are many more made that are not explicitly stated. In Sec. 3.1, it is discussed that $P_{\theta}(\tilde{Y} \mid X)$ can act as a surrogate for $P(Y \mid X)$, but there is no formal explanation on what this means. There is also little justification on the implications of the clustering algorithm. The results of this approach heavily depend on the outputs of the clustering algorithm, so there must be some implicit assumption that the clustering algorithm outputs something relevant to the causal structure of the dataset, which should be explicitly stated.

Given these concerns, I cannot recommend the paper for acceptance in its current form. I am open to hearing author responses in case I misunderstood something.

EDIT: Following rebuttal, I am raising my score from 3 to 6.

**Questions:**

1. In the introduction, it is mentioned that the causal graphs are assumed to be acyclic. However, there seems to be a cycle in Fig. 1b from $X_2 \rightarrow X_d \rightarrow Y \rightarrow X_2$. This seems to be a contradiction, could the authors clarify on this point?

2. In Sec. 3.1, it is mentioned that $P(\tilde{Y} = \tilde{y} \mid Y’ = y’, X = x)$ should equal $1 / C$ for each $x$. Does this only hold if the distribution of labels is uniform?

---

> ### Author Response · Authors · 2023-11-16
> **Rebuttal by Authors (Part 1)**
>
> **Q1. It is not formally explained what it means for a distribution to “inform” another, yet this seems to be key to understanding the proposed approach.**
>
> **A1.** Thank you for pointing this out. The word "inform" is taken from the definition of the principle of independent mechanisms. Specifically, on Page 19 of the book "Elements of Causal Inference: Foundations and Learning Algorithms", it states:
>
> "(Independent mechanisms) The causal generative process of a system’s variables is composed of autonomous modules that do not inform or influence each other. In the probabilistic case, this means that the conditional distribution of each variable given its causes (i.e., its mechanism) does not inform or influence the other conditional distributions. In the case of only two variables, this reduces to independence between the cause distribution and the mechanism producing the effect distribution."
>
> **An Example** To concretely explain that the conditional distribution of each variable, given its causes (i.e., its mechanism), does not inform or influence the other conditional distributions, let"s consider an interesting example that follows the generative process of causal datasets.
> -   We act as the data collector.  1). we randomly sample a photo $\boldsymbol{X}$ from Instagram.
> -  Let Tom be the annotator. He will annotate each $\boldsymbol{X}$ we pass but without any knowledge of $P(\boldsymbol{X})$.
> -  Following the generative process, 2). we pass the photo $\boldsymbol{X}$ to Tom. Tom writes the label $Y$ on the back of the photo $\boldsymbol{X}$ and puts the photo in a black box.
> -  We repeat the process 1), and Tom repeats the process 2).
>
> The question then arises: **can we act like a clustering algorithm by looking at $P(\boldsymbol{X})$ to understand how photos in the box are labeled?**
>
> Generally, the answer is no. Intuitively, there are too many possible ways to annotate the photo. Tom could label the photos based on whether the image contains a human, the number of humans, night vs. day, and other rules. We have no idea about his mechanism by only looking at $P(\boldsymbol{X})$. In this case, $P(\boldsymbol{X})$ does not inform $P(Y|\boldsymbol{X})$.
>
> **We have added the above example to Appendix D of the revised version to help readers understand. Please let us know if it is clear now.**
>
> **Q2. It seems that the assumption of no latent confounder has not been mentioned in the paper but is actually key to the effectiveness of the proposed approach.**
>
> **A2.** The latent confounder assumption is discussed in footnote 1 of the original version. We have revised our paper to make it clearer and emphasize it in the main text as follows:
>
> ________________________________
> > We assume that there are no latent confounders, similar to many existing causal discovery methods. If latent confounders exist, our method will interpret it as anticausal, as in such cases, $P(\boldsymbol{X})$ also contains information about $P(Y|\boldsymbol{X})$, resembling an anticausal case. To further check whether it is an anticausal or confounded case, existing methods specifically designed for finding latent confounders can be applied (Chen et al., 2022; Huang et al., 2022).
> ________________________________
>
>
> **Q3. The conclusion that “the conditional distributions of each variable, given all causal parents, are independent entities that do not share any information” only holds under the Markovianity assumption (i.e. no unobserved confounders) but not independent mechanisms.**
>
>
> **A3.** We are sorry for the confusion. Our aim here is to explain the principle of independent mechanisms, which does not directly relate to the Markovianity assumption. This principle works under the assumption that all causal variables and underlying causal structures are given, thereby preventing the existence of latent variables.
>
> We have revised our paper to highlight this under the paragraph "The Principle of Independent Mechanisms" as follows:
> ________________________________
> > In the probabilistic cases (detailed in Chapter 2 of Peters et al. (2017)), the principle states that "the conditional distribution of each variable given its causes (i.e., its mechanism) does not inform or influence the other conditional distributions." In other words, assuming all underlying causal variables are given and there are no latent variables, the conditional distributions of each variable, given all its causal parents (which can be an empty set), do not share any information and are independent of each other.  To explain the independence concretely, we include an example in Appendix D.
> ________________________________
>
>
> However, in practice, our method does require the assumption of no latent confounders. Accordingly, we have emphasized this point in our paper (refer to **A2**).

---

> ### Author Response · Authors · 2023-11-16
> **Rebuttal by Authors (Part 2)**
>
> **Q4. Is the paper claiming to somehow infer the data-generating mechanisms from the distributions?**
>
> **A4.** We focus on checking whether the dataset is causal or anticausal without latent confounders. It is directly related to the data generative process. If the dataset is causal, $Y$ must not be a cause of some variables in $\boldsymbol{X}$, if the dataset is anticausal, then $Y$ must be an effect of $\boldsymbol{X}$.
>
>
> **Q5. The approach seems to rely on the property that within causal datasets, observed labels are evenly distributed among the clusters of P(\boldsymbol{X}), while they are not in anticausal datasets. This property does not seem to be related to any causal properties.**
>
>
> **A5.** We do not require any additional assumptions to assume that observed labels are evenly distributed among the clusters of $P(\boldsymbol{X})$.
>
> Intuitively the logic is that, if the independent mechanisms hold true, as in the example mentioned in **A1**, for a causal dataset, only “looking” at $P(\boldsymbol{X})$ tells us nothing about the clean class posterior $P(Y|\boldsymbol{X})$. Moreover, since the posterior for the observed label $P_\theta(\tilde{Y}|\boldsymbol{X})$ can be seen as an estimation of $P(Y|\boldsymbol{X})$ with some error, if $P(\boldsymbol{X})$ reveals nothing about $P(Y|\boldsymbol{X})$, it also reveals nothing for $P_\theta(\tilde{Y}|\boldsymbol{X})$.
>
> Furthermore, it is important to note that if a dataset contains **clusters** where the labels within each cluster are similar, then we can directly conclude that $P(\boldsymbol{X})$ contains information about $P_\theta(\tilde{Y}|\boldsymbol{X})$. Therefore, such a dataset is generally an anticausal dataset.
>
>
>
>
> **Q6. The distribution $P(\boldsymbol{X},Y)$  can be factorized as either Eq. 2 or Eq. 3 regardless of causal orientation. It is not clear what is the causal consequence of choosing one over the other.**
>
>
> **A6.** We apologize for any confusion caused. Our intention is not to favor one factorization over the other. We aim to emphasize an asymmetric property that
> - In a causal dataset, $P(\boldsymbol{X})$ and $P(Y|\boldsymbol{X})$ align with the causal direction, meaning that $P(\boldsymbol{X})$ and $P(Y|\boldsymbol{X})$ are independent and do not share information.
>
> - In an anticausal dataset, $P(\boldsymbol{X})$ and $P(Y|\boldsymbol{X})$ do not align with the causal direction. As a result, $P(\boldsymbol{X})$ and $P(Y|\boldsymbol{X})$ are not independent and may share information.
>
>
> We guess that the confusion comes from the causal factorization and Fig 1. We have made adjustments to our revised version. It would be very appreciated if you could further provide some suggestions, and kindly let us know whether this is clear now.
>
> The adjustments are as follows:
>
> - Remove Eq. (2) and Eq. (3); describe the dependence relation with language. Eq. (2) and Eq. (3) in our original version have not fully factorized all variables according to the causal direction. We think this can cause confusion. Fully factorizing this results in a complex and unwieldy form, which does not help in understanding our intention. Our intention is to explain the conditions under which $P(X)$ contains information about $P(Y|X)$. Specifically, we change this part to:
>
> _________________________
> >We follow the definition of Causal and Anticausal datasets from (Scholkopf et al., 2012).
> For a causal dataset, *some variables in $\boldsymbol{X}$ act as causes for the class $Y$, and no variable in $\boldsymbol{X}$ is an effect of the class $Y$ or shares a common cause with the class $Y$* (e.g., Fig. 1a). In this case, $Y$ can only be an effect of some variables in $\boldsymbol{X}$. Two distributions $P(\boldsymbol{X})$ and $P(Y|\boldsymbol{X})$ satisfy the independent causal mechanisms. The distribution $P(\boldsymbol{X})$ does not contain information about $P(Y|\boldsymbol{X})$.
>
> > For anticausal datasets, however, the label $Y$ can be a cause variable. In such cases, the independent causal mechanisms are not satisfied for $P(\boldsymbol{X})$ and $P(Y|\boldsymbol{X})$, implying that $P(\boldsymbol{X})$ contains information about $P(Y|\boldsymbol{X})$.
> _________________________
>  - We have also updated the Fig. 1 to ensure that the cyclic graph case does not occur.

---

> ### Author Response · Authors · 2023-11-16
> **Rebuttal by Authors (Part 3)**
>
> **Q7. There seems to be a cycle in Fig. 1b.**
>
> **A7.** Thank you very much for highlighting this issue. You are correct. We have removed the edge from $X_2$ to $X_4$ to ensure the graph is acyclic, and we have incorporated these changes in our revised version. Please kindly let us know if the revisions are now clear and adequately address your concerns.
>
>
>
> **Q8. In Sec. 3.1, it is mentioned that $P(\tilde{Y}=\tilde{y}|Y’=y’,X=x )$ should equal $1/C$  for each  $\boldsymbol{X}$. Does this only hold if the distribution of labels is uniform?**
>
> **A8.** That is not the case. Our goal is to explain the specific condition that will be met if $P(\boldsymbol{X})$ does not contain information about $P(Y|\boldsymbol{X})$. This condition is that $P(\tilde{Y}=\tilde{y}|Y'=Y', X=x) = 1/C$. It indicates that predictions made by exploiting $P(\boldsymbol{X})$ are essentially just random guesses of the observed label $\tilde{Y}$.
>
> In practice, this probability is hard to estimate. Our estimator does not estimate it. Instead, we have designed a noise injection approach. For a detailed explanation of the rationale behind the noise injection, please refer to our response to Reviewer 32x9 in**A5** and kindly let us know if anything is unclear.
>
>
>
>
>
> **Q9. Assumptions are unclear.**
>
> **A9.** Thank you for pointing this out. Our assumptions are discussed in Appendix F in the original version. We have moved the assumptions to the main text (as also suggested by Reviewer aduf).
> **Note that we have also justified the implication of the clustering algorithm.**
>
> Specifically, to accommodate the changes, we have moved the paragraph "Causal Graphs and Structural Causal Models (SCM)" to the Appendix. We have also adjusted the "Discussion & Limitation" section, now placing it in the main text at the end of the "Theoretical Analyses" section. The specific adjustments are as follows:
> ________________________
> > **Assumptions for Discovering Causal and Anticausal Relationships**
> Our method is based on commonly accepted assumptions in causal discovery: causal faithfulness, acyclic graph assumption, absence of latent confounders, and independent causal mechanisms  (Peters et al., 2014).
> To ensure that the disagreements (or expected risks) under different noise levels remain constant in a causal setting when employing RoCA, an additional assumption is needed to constrain the types of label errors in datasets.
> Specifically, this assumption states that for every instance and clean class pair $(x, y)$, the observed label $\tilde{y}$ is derived with a noise rate $\rho_x$ such that $P(\tilde{Y}=\tilde{y}|Y=y,\boldsymbol{X}=x)=\frac{\rho_{x}}{C-1}$ for all $\tilde{y} \neq y \land \tilde{y} \in C$. This assumption is applicable not only when data contains instance-dependent label errors but also when there are no label errors or when data contains class-dependent errors \cite{patrini2017making, xia2019anchor, li2021provably}.
>
> > Furthermore, to use $P_{\theta}(\tilde{Y}|\boldsymbol{X})$ as a surrogate for $P(Y|\boldsymbol{X})$, we assume a dependence between $P_{\theta}(\tilde{Y}|\boldsymbol{X})$ and $P(Y|\boldsymbol{X})$. This assumption usually holds, as the absence of such a dependence would imply that the annotation mechanism $P_{\theta}(\tilde{Y}|\boldsymbol{X})$ is just a random guess of $P(Y|\boldsymbol{X})$, making the observed label $\tilde{Y}$ meaningless.
>
> > Additionally, the effectiveness of our method can be influenced by the choice of a backbone clustering method. Specifically, when handling an anticausal dataset, our approach depends on a clustering method that is capable of extracting relevant information from $P(\boldsymbol{X})$ to predict $P(Y|\boldsymbol{X})$, rather than simply guessing randomly. Owing to recent advances in unsupervised and self-supervised methods, some approaches (Niu et al., 2021) based on contrastive learning have achieved performance competitive with supervised methods on benchmark image datasets like CIFAR10 and MNIST.
> ________________________

---

> ### Author Response · Authors · 2023-11-16
> **Rebuttal by Authors (Part 4)**
>
> **Q10. In Sec. 3.1, it is discussed that $P_\theta(\tilde{Y}|\boldsymbol{X})$ can act as a surrogate about $P(Y|\boldsymbol{X})$, but there is no formal explanation of what this means.**
>
> Note that we consider the possibility of label errors in the data. In this scenario, the clean class $Y$ is latent, and the observed labels $\tilde{Y}$ contain errors.
>
> The distribution $P_\theta(\tilde{Y}|\boldsymbol{X})$ can serve as a surrogate for $P(Y|\boldsymbol{X})$. This means that to determine whether a dataset is causal or anticausal, instead of checking whether $P(\boldsymbol{X})$ contains information about $P(Y|\boldsymbol{X})$, we can examine whether $P(\boldsymbol{X})$ contains information about $P_\theta(\tilde{Y}|\boldsymbol{X})$. Therefore, $P_\theta(\tilde{Y}|\boldsymbol{X})$ effectively acts as a proxy for $P(Y|\boldsymbol{X})$.
>
> The underlying assumption is that the distributions $P_\theta(\tilde{Y}|\boldsymbol{X})$ and $P(Y|\boldsymbol{X})$ are dependent. In other words, $\tilde{Y}$ is meaningful and should be an estimation of $Y$, rather than merely a random guess. This assumption is generally reasonable since the observed labels are usually not completely random.
>
> Mathematically, this relationship can be expressed in terms of conditional probabilities.
>
> - If $P_\theta(\tilde{Y} |\boldsymbol{X})$ and $P_\theta(Y |\boldsymbol{X})$ are completely independent, then for every instance $\boldsymbol{X}$, the probability $P(\tilde{Y}=\tilde{y} |Y =y , \boldsymbol{X}= \boldsymbol{x})$ equals $1/C$, where $C$ is the number of classes. This implies that for each instance $\boldsymbol{X}$, its observed label $\tilde{Y}$ is essentially a random guess of its clean class $Y$.
>
> - However, if $P_\theta(\tilde{Y} |\boldsymbol{X})$ and $P_\theta(Y |\boldsymbol{X})$ are dependent, then typically, $P(\tilde{Y}=\tilde{y} |Y =y , \boldsymbol{X}= \boldsymbol{x}) \neq 1/C$. In such cases, the observed label $\tilde{Y}$ provides information about the clean class $Y$, indicating a non-random relationship between the observed and true labels.

---

> ### Author Response · Authors · 2023-11-21
> **Dear Reviewer g7sE, are there any further concerns, thanks**
>
> Dear Reviewer g7sE,
>
> Thank you for your efforts in reviewing our paper and for pointing out the cycle case in Figure 1(b). We have revised our paper accordingly, including *making our assumptions clear, emphasizing confounded cases, and revising the preliminaries and experimental settings*.
>
> Please kindly let us know if any explanations remain unclear; we will gladly provide further clarification.
>
> Warm regards,
>
> Authors

---

> ### Author Response · Authors · 2023-11-22
> **Rolling discussion coming to an end – awaiting your valuable feedback**
>
> Dear Reviewer g7sE,
>
> Thank you once again for your invaluable contributions to reviewing our paper.
>
> Considering the impending discussion deadline, we respectfully invite your prompt response. If you require any further details or seek clarification on specific aspects of the paper, please kindly let us know.
>
> Warm regards,
>
> Authors

---

> ### Comment · Reviewer_g7sE · 2023-11-22
> **RE: Rebuttal**
>
> Thank you for the clarifications, they have dramatically improved my understanding of the work. The edits are appreciated. The addition of the assumptions paragraph that you have mentioned makes the work significantly more concrete. For these reasons, I will raise my score to a 6.

---

> > ### Author Response · Authors · 2023-11-22
> > **Thanks**
> >
> > Dear Reviewer g7sE,
> >
> > Thank you so much for your valuable contributions to our paper. This achievement would not have been possible without your efforts.
> >
> > Sincerely,
> >
> > Authors

---

### Official Review · Reviewer_32x9 · 2023-11-01

**Soundness:** 2 fair
**Presentation:** 2 fair
**Contribution:** 2 fair
**Rating:** 3
**Confidence:** 3

**Summary:**

This paper aims to discern if a data generation process leans towards being causal or anti-causal. Introducing the Robust Causal and Anticausal (RoCA) Estimator, the authors attempt to differentiate the two by investigating if the instance distribution, $P(X)$, offers pertinent details about the prediction task, $P(Y|X)$. They opted for the noisy class-posterior distribution, $P(\tilde{Y}|X)$, to act as a stand-in for $P(Y|X)$, and devised clusters using unsupervised or self-supervised techniques. Their findings suggest that in a causal scenario, there's no correlation between mismatch and noise levels, while in an anti-causal context, a correlation exists. The paper furnishes empirical evidence to support these claims.

**Strengths:**

1. The problem of interest is an important topic in casual discovery.

2. The method proposed overall sounds interesting and new.

3. The paper is well written.

**Weaknesses:**

1. The core premise of the paper, notably the logic of employing \(p(x)\) predictiveness to discern between causal and anti-causal directions for \(p(y | x)\), lacks solid substantiation. A deeper justification, supported by empirical data, would strengthen this assumption.

2. The paper does not present clear identifiability results. The claim that it's unnecessary to identify all potential causal relationships among single-dimensional variables is made without sufficient exploration. Additionally, concerns arise in an anti-causal context with a potential cyclic graph, questioning whether \(P(X)\) indeed offers valuable insight for the prediction task \(P(Y|X)\).

3. There seems to be a discrepancy in the paper's foundational assumptions on causal inference. While the authors state they align with the definitions in Sch¨olkopf et al. (2012), their handling of the Anticausal definition, especially regarding confounded cases, suggests otherwise. The paper needs to clarify its stance on unmeasured confounders.

4. The methodology for determining the noisy distribution and constructing \(P(\tilde{Y}|X)\) appears to lack a clear rationale. Offering detailed reasons for the "Noise Injection" approach and possibly introducing sensitivity analysis would bolster this section.

5. The method's practical relevance raises concerns. While the novel concept of integrating the causal direction into (semi-)supervised problems is compelling, its adoption in real-world applications remains questionable.

6. The paper seems to omit a comprehensive review of the related literature. Engaging more deeply with existing academic contributions would provide readers with valuable context, facilitating a better understanding of the paper's novelty and its positioning in the wider domain.

7. The overall organization and clarity of the paper need improvement. The section discussing experiments is notably intricate, making navigation challenging. A clearer structure and presentation would significantly improve the paper's readability.

**Questions:**

Please consider addressing the weakness I mentioned above.

---

> ### Author Response · Authors · 2023-11-16
> **Rebuttal by Authors (Part 1)**
>
> **Q1. Why can the predictiveness of exploiting $P(\boldsymbol{X})$ in relation to $P(Y|\boldsymbol{X})$ can discern between causal and anti-causal directions?  Need explanation and empirical support.**
>
> **A1.** This is immediately from the principle of independent mechanisms. In brief, if exploiting $P(\boldsymbol{X})$ is useful for help learning labels, then it implies that  $P(\boldsymbol{X})$ contains information about $P(Y|\boldsymbol{X})$. This holds for anticausal cases but not causal cases in general.
>
> (refer to “On Causal and Anticausal Learning”, Page 2 https://arxiv.org/ftp/arxiv/papers/1206/1206.6471.pdf, and “Semi-supervised learning, causality, and the conditional cluster assumption”, Page 1, https://arxiv.org/pdf/1905.12081.pdf).
>
> _______________________________
> Here is a **detailed explanation for those who may be interested**.
>
> Specifically, on Page 19 of the book "Elements of Causal Inference: Foundations and Learning Algorithms", it states:
>
> "(Independent mechanisms) The causal generative process of a system’s variables is composed of autonomous modules that do not inform or influence each other. *In the probabilistic case, this means that the conditional distribution of each variable given its causes (i.e., its mechanism) does not inform or influence the other conditional distributions…"*
>
> - In a causal dataset, as defined, $Y$ is an effect of some variables in instances $\boldsymbol{X}$. Under this scenario, independent mechanisms hold true for the distribution $P(\boldsymbol{X})$ and $P(Y | \boldsymbol{X})$. Therefore, $P(\boldsymbol{X})$ does not contain information about $P(Y | \boldsymbol{X})$. This implies that exploiting $P(\boldsymbol{X})$ to predict the class $Y$ would yield accuracy equivalent to random guessing (Please also refer to the concrete example mentioned in **A1** of our rebuttal to Reviewer g7sE).
>
> - Conversely, in an anticausal dataset, $Y$ is a cause of some variables in $\boldsymbol{X}$. Examining $P(Y | \boldsymbol{X}) = P(Y | \{X_1, X_2, \dots, X_d\})$, we find that some effects of $Y$ are included in the condition, which contradicts the independent mechanisms. This contradiction implies that the instance distribution $P(\boldsymbol{X})$ contains information about $P(Y | \boldsymbol{X})$. Consequently, the distribution $P(\boldsymbol{X})$ is useful for predicting the class $Y$ in anticausal scenarios.
>
> This is why the predictiveness of $P(\boldsymbol{X})$ is crucial for discerning between causal and anti-causal directions for $P(Y | \boldsymbol{X})$.
> _______________________________
>
> **The empirical support is directly shown in our synthetic experiments (see Fig. 4).** Intuitively, by exploiting $P(\boldsymbol{X})$, we can get some learned cluster IDs. These cluster IDs can then be mapped to pseudo labels. This mapping is based on the majority of observed labels that share the same cluster IDs.
> - Under causal cases, injecting different levels of label errors adds randomness to observed labels, but the disagreement between pseudo labels and observed labels does not change. It implies that pseudo labels are independent of observed labels and essentially random predictions.  Therefore, $P(\boldsymbol{X})$ can not help learn $P(Y|\boldsymbol{X})$.
>
> -  Under anticausal cases, injecting different levels of errors causes the disagreement between pseudo labels to change, indicating that learned pseudo labels are better than random guesses of observed labels. After adding noise to observed labels, this weakens the dependence between pseudo labels and observed labels.
>
>
> **Empirical Support In Previous Work** The paper ”On Causal and Anticausal Learning” shows that If the dataset is causal, then the performance of semi-supervised learning is limited (refer to Fig. 6, 7 and 8 in their paper). If the dataset is anti-causal, the performance of semi-supervised learning improves. It is because semi-supervised learning relies on exploiting $P(\boldsymbol{X})$ to help learn $P(Y|\boldsymbol{X})$. If it is causal, $P(\boldsymbol{X})$ can not be used to help predict $P(Y|\boldsymbol{X})$ in general.
>
>
>
> **Q2. (Regarding Identifiability) The claim that it is unnecessary to identify all potential causal relationships among single-dimensional variables is made without sufficient exploration.**
>
> **A2.** To detect causal or anticausal relations, the objective is to check the causal association between $Y$ and $\boldsymbol{X}$. **Specifically, We are interested in knowing whether $Y$ is a cause of $\boldsymbol{X}$**.
> Identifying causal associations between pairs of single-dimensional variables in the instance $\boldsymbol{X}$ only can tell us whether a single-dimensional variable $X_i$ is a cause of a single-dimensional variable $X_j$, where $X_i, X_i \in \boldsymbol{X}$. Detecting these causal associations is not our interest at all. Therefore, determining all potential causal relationships among single-dimensional variables is not a requirement of our approach.

---

> ### Author Response · Authors · 2023-11-16
> **Rebuttal by Authors (Part 2)**
>
> **Q3. (Regarding Identifiability) Concerns arise in an anti-causal context with a potential cyclic graph.**
>
> **A3.** Thank you for pointing this out. In the introduction, we mentioned that our method requires the acyclic graph assumption, a common assumption used by most causal discovery methods (e.g., PC, GES, LinGAM, FCI, etc.).
>
> We think that the confusion comes from the causal factorization and Fig 1. We have made adjustments to our revised version. **It would be very appreciated if you could further provide some suggestions, and kindly let us know whether this is clear now.**
>
> The adjustments are as follows:
>
> - Remove  Eq. (2) and Eq. (3); describe the dependence relation with language. Eq. (2) and Eq. (3)  in our original version have not fully factorized all variables according to the causal direction. We think this can cause confusion. Fully factorizing this results in a complex and unwieldy form, which does not help in understanding our intention. Our intention is to explain the conditions under which $P(X)$ contains information about $P(Y|X)$. Specifically, we change this part to:
>
> _________________________
> > We follow the definition of Causal and Anticausal datasets from (Scholkopf et al., 2012).
> For a causal dataset, *some variables in $\boldsymbol{X}$ act as causes for the class $Y$, and no variable in $\boldsymbol{X}$ is an effect of the class $Y$ or shares a common cause with the class $Y$* (e.g., Fig. 1a). In this case, $Y$ can only be an effect of some variables in $\boldsymbol{X}$. Two distributions $P(\boldsymbol{X})$ and $P(Y|\boldsymbol{X})$ satisfy the independent causal mechanisms. The distribution $P(\boldsymbol{X})$ does not contain information about $P(Y|\boldsymbol{X})$.
>
> > For anticausal datasets, however, the label $Y$ can be a cause variable. In such cases, the independent causal mechanisms are not satisfied for $P(\boldsymbol{X})$ and $P(Y|\boldsymbol{X})$, implying that $P(\boldsymbol{X})$ contains information about $P(Y|\boldsymbol{X})$.
> _________________________
> We have also updated the Fig. 1 to ensure that the cyclic graph case does not occur.
>
>
>
> **Q4. The paper needs to clarify its stance on unmeasured confounders.**
>
> **A4.** The latent confounders are mentioned in footnote 1 of the original version. We have revised our paper to make it clearer and emphasize it in the main text as follows.
>
> ________________________________
> > We assume that there are no latent confounders, similar to many existing causal discovery methods. If latent confounders exist, our method will interpret it as anticausal, as in such cases, $P(\boldsymbol{X})$ also contains information about $P(Y|\boldsymbol{X})$, resembling an anticausal case. To further check whether it is an anticausal or confounded case, existing methods specifically designed for finding latent confounders can be applied (Chen et al., 2022; Huang et al., 2022).
> ________________________________

---

> ### Author Response · Authors · 2023-11-16
> **Rebuttal by Authors (Part 3)**
>
> **Q5. Need clear rationale for determining the noisy distribution and constructing $P(\tilde{Y}^\rho |\boldsymbol{X})$. Offering detailed reasons for the "Noise Injection" approach.**
>
> **A5.** **We would like to first provide some background for a good self contain then answer the rationale.**
>
>
> ___________________
> **Backgound**
>
> Reminding readers that the core idea of our method, as detailed in **A1**, is to check whether $P(\boldsymbol{X})$ can help predict $P(Y|\boldsymbol{X})$ and thus determine causal and anticausal relationships.
>
> Considering the possibility of label errors in the data, the clean class $Y$ becomes latent. Instead, we have observed labels $\tilde{Y}$ that contain errors. Therefore, instead of checking whether $P(\boldsymbol{X})$ contains information about $P(Y|\boldsymbol{X})$, **we check if $P(\boldsymbol{X})$ contains information about $P_\theta(\tilde{Y}|\boldsymbol{X})$.** Since $P_\theta(\tilde{Y} |\boldsymbol{X})$ can be viewed as an estimation of $P(Y |\boldsymbol{X})$ with errors, if $P(\boldsymbol{X})$ can help predict the posterior of observed labels $P_\theta(\tilde{Y} |\boldsymbol{X})$, it can also aid in predicting the posterior of classes $P(Y |\boldsymbol{X})$.
>
> Let $Y'$ be the pseudo label obtained by exploiting $P(\boldsymbol{X})$. (By exploiting $P(\boldsymbol{X})$, we can derive learned cluster IDs, which are then mapped to pseudo labels based on the majority of observed labels sharing the same cluster IDs.)
> ___________________
>
>
> If $P(\boldsymbol{X})$ does not contain relevant information, the pseudo label $Y'$ is just a random guess of the observed $\tilde{Y}$ for every instance $\boldsymbol{X}$, i.e., $P(\tilde{Y}|Y',X)$ equals $1/C$ for every $\boldsymbol{X}$. **However, directly and accurately estimating this probability is challenging since a particular instance of $\boldsymbol{X}$ might appear only once in the dataset.**
>
> **Using a noise injection approach to avoid estimating $P(\tilde{Y}|Y',X)$.**
> The underlying rationale is that
> - if $P(\boldsymbol{X})$ does not include information about $P(\tilde{Y}|\boldsymbol{X})$, then the pseudo label $Y'$ obtained by exploiting $P(\boldsymbol{X})$ should always be a random guess of $\tilde{Y}$ when random noise is added.
> - However, if $P(\boldsymbol{X})$ contains information about $P(\tilde{Y}|\boldsymbol{X})$, increasing the noise level in $\tilde{Y}$ will weaken the dependency relationship, introducing more uncertainty and randomness, making $\tilde{Y}$ more unpredictable.
> - To formalize this, we propose designing a hypothesis test aimed at determining whether the trend of $P(Y'|\tilde{Y}^\rho,\boldsymbol{X})$ remains flat (indicating no information content in $P(\boldsymbol{X})$ about $P(\tilde{Y}|\boldsymbol{X})$) or shows significant changes (indicating the presence of information) as the noise level $\rho$ increases.
>
> **Designing the injected noise distribution to fulfill theoretical robustness.**
> It is also important to note that proving this claim theoretically requires an additional necessary condition: the probability of flipping an observed label to any other class for each instance should be $\frac{\rho_x}{C-1}$. If this condition is not met, the distribution $P(\tilde{Y}^\rho|Y’,\boldsymbol{X})$ will change with different levels of noise injection. To ensure theoretical robustness, our noise injection method is designed to fulfill this condition.
>
>
>
> **Q6. Introducing sensitivity analysis would bolster this section.**
>
> **A6.** Our estimator does not rely much on hyperparameters. It is designed to use a hypothesis test to determine causality and anticausality. The noise level $\rho$ is not a hyperparameter; instead, it is randomly sampled for the hypothesis test. The significant level is set to 0.05 which is a common threshold for hypothesis tests. If you are interested in any sensitivity analysis using backbone clustering methods, please let us know. We are more than willing to conduct additional experiments and report the results.

---

> ### Author Response · Authors · 2023-11-16
> **Rebuttal by Authors (Part 4)**
>
> **Q6. Introducing sensitivity analysis would bolster this section**
> **A6.** Our estimator does not rely much on hyperparameters. It is designed to use a hypothesis test to determine causality and anticausality. The noise level $\rho$ is not a hyperparameter; instead, it is randomly sampled for the hypothesis test. The significant level is set to 0.05 which is a common threshold for hypothesis tests. If you are interested in any sensitivity analysis using backbone clustering methods, please let us know. We are more than willing to conduct additional experiments and report the results.
>
>
>
> **Q7. The method's practical relevance raises concerns. Its adoption in real-world applications remains questionable.**
>
> **A7.** It is important to note that the application of our method goes beyond just determining semi-supervised learning. As highlighted in the last paragraph of the introduction,**our method can also be utilized to detect the causal direction between a set of continuous (or discrete) variables and a discrete variable.** This is significant, as many observed variables in datasets are discretized. In our experiments, we have included a variety of real-world datasets, such as *Breastcancer*, *Splice* (DNA sequences), and WDBC (tumor features). There are numerous other potential applications in different areas. Here, we provide two examples:
>
> - **Healthcare and Medicine:** Determining whether lifestyle factors (such as exercise duration or dietary habits - continuous variables) lead to specific health outcomes (like developing diabetes or heart disease - discrete variables), or vice versa. For example, understanding if increased exercise (a continuous variable) reduces the risk of heart disease (a discrete variable), rather than heart disease risk influencing exercise patterns, can significantly influence treatment and prevention strategies.
>
> - **Environmental Science:** Determining whether environmental conditions (like pollution levels - continuous variables) cause specific ecological events (such as the occurrence of acid rain - a discrete variable) or if the relationship is the other way around. Determining, for instance, whether higher pollution levels (a continuous variable) are the cause of increased occurrences of acid rain (a discrete variable), rather than acid rain occurrences affecting pollution levels, is vital for effective environmental policy-making and intervention strategies.
>
> **Q8. The paper seems to omit a comprehensive review of the related literature.**
>
> **A8.** It has been provided in supplementary due to the limited space. We have also emphasized at the start of section 2  "Owing to space limitations, a review of existing causal discovery methods is left in Appendix B."

---

> ### Author Response · Authors · 2023-11-16
> **Rebuttal by Authors (Part 5)**
>
> **Q9. The section discussing experiments is notably intricate.**
>
> Thank you very much for your suggestion. We apologize for any inconvenience caused by the compression of the experiments section due to space limitations. We have made our best effort to include more details:
>
> 1. To clarify how the p-value is obtained in Tables, we have shifted and revised the discussion from the Method section to the Experiments section. This allows readers to quickly grasp the main idea behind the p-value calculation. The revised section is as follows:
> __________________
> > To rigorously validate the disagreement, rather than directly evaluating if the slope $\hat{\beta}_1$ contained from Eq.  (4) is near $0$, we use a hypothesis test on the slope.
> The level of noise $\rho$ is randomly sampled $20$ times from a range between $0$ and $0.5$. For every selected noise level, a disagreement between $Y'$ and $\tilde{Y}$ can be calculated. Consequently, a slope value is calculated from the correlation of these $20$ noise levels and their respective disagreement ratios. By repeating such a procedure for $30$ times, a set of slope values can be obtained. These slope values are then utilized in our hypothesis test to verify if the average slope is significantly different from $0$. The details of this test are provided in Appendix A. Intuitively, if the resulting $p$-value from the test exceeds $0.05$, the slope is likely to close to zero, indicating a causal dataset. Conversely, a $p$-value below $0.05$ suggests an anticausal dataset.
>
> __________________
>
>
> 2. We have revised paragraph Disagreements with Different Noise Levels on Synthetic Datasets and Disagreements with Different Noise Levels on Synthetic Datasets to contain as much information under limited space as follows.
> __________________
> >**Disagreements with Different Noise Levels on Synthetic Datasets**
> Fig. 4 demonstrates the trend of disagreement with different noise levels for *synCausal* and *synAnticausal* datasets. To construct datasets with label errors,  $30\%$ label errors are added into these datasets. For the *synCausal* dataset, the trend of disagreement remains unchanged at $0.5$ with the increase of injected noise rates, and the slope $\hat{\beta}_1$ of the regression line is close to $0$. This is because $Y'$ is poorly estimated and should be a random guess of noised $\tilde{Y}"$.
>
> > On the other hand, for the *synAnticausal* dataset with small label errors (e.g., Sym and Ins-$10\%$ to $20\%$), there is a strong positive correlation between the disagreement and the noise level. In this case, $Y'$ is better than a random guess of both $\tilde{Y}$ and the latent clean class $Y$. Specifically, with the increase of noise level $\rho$, the corresponding $\tilde{Y}^{\rho}$ becomes more seriously polluted and tends to deviate far away from the observed label $\tilde{Y}$. This results in a larger disagreement between $\tilde{Y}^{\rho}$ and $Y'$.
>
> >**Performance of RoCA on Real-World Datasets**
> We have also benchmarked the RoCA method against other causal discovery algorithms. Our results, as presented in Tab. 1 and Tab. 2, demonstrate that our method is both more accurate and robust. In these tables, the term "unknown" indicates cases where the algorithm either failed to detect the causal relation, or did not complete the analysis within a feasible running time. Note that only RoCA can applied to image datasets CIFAR10N and Clothing1M to detect causal and anticausal relations.
> ________________

---

> > ### Author Response · Authors · 2023-11-23
> > **Dear Reviewer 32x9, Rolling discussion coming to an end – thank you and awaiting your feedback**
> >
> > Dear Reviewer 32x9,
> >
> > Thank you for your hard work. Your advice is valuable to us.
> >
> > As the rolling discussion period for our paper draws to a close, we are reaching out to humbly remind you of our eagerness for your feedback. Your feedback is important in helping us address any outstanding concerns and ensuring a smooth review process.
> >
> > Many thanks,
> >
> > Authors

---

> ### Author Response · Authors · 2023-11-21
> **Dear Reviewer 32x9, please let us know if you have any further concerns, thanks**
>
> Dear Reviewer 32x9,
>
> Thank you for your efforts in reviewing our paper. We have revised the paper according to your constructive comments.
>
> Please kindly let us know If any explanations remain unclear. we will gladly provide further clarification.
>
> Warm regards,
>
> Authors

---

> ### Author Response · Authors · 2023-11-22
> **Rolling discussion coming to an end –  awaiting your valuable feedback**
>
> Dear Reviewer 32x9,
>
> As the rolling discussion period for our paper is coming to a close, we are still waiting for your feedback. If you need further clarification on any aspect of the paper, please do not hesitate to contact us. We are more than happy to address any questions or concerns you may have to ensure a smooth review process.
>
> Warm regards,
>
> Authors

---

> ### Author Response · Authors · 2023-11-23
> **Dear Reviewer 32x9, a summarize of our response, and do you have further concerns?**
>
> Dear Reviewer 32x9,
>
> As the rolling discussion period for our paper nears its end, we want to gently remind you that we are eagerly anticipating your valuable feedback.
>
> To facilitate a quicker understanding, we have prepared a brief summary of our response:
>
> 1. **Predictiveness of $P(\boldsymbol{X})$ for Causal and Anticausal Directions**: This is directly from the principle of independent mechanisms. We show both theoretical and empirical support from existing work.
> 2. **Necessity of Identifying Causal Relationships Among Single-Dimensional Variables**:
> To detect causal or anticausal relation, we are only interested in knowing whether $Y$ is a cause of $\boldsymbol{X}$ instead of knowing whether a single-dimensional variable $X_i$ is a cause of a single-dimensional variable $X_j$, where $X_i, X_i \in \boldsymbol{X}$.
> 3. **Potential Cyclic Graph**: We adjusted our manuscript to emphasize the acyclic graph assumption and clarified causal factorization.
> 4. **Latent Confounders**: We emphasize latent confounders in the main text, explaining our assumption of their non-existence.
> 5. **Rationale for Noise Injection**: We explained the 1). rationale of noise injection; 2). choice of the injected noise distribution; 3). our hypothesis test in noise injection.
> 6. **Sensitivity Analysis**: We highlighted the limited reliance of our estimator on hyperparameters, with the noise level $\rho$ being randomly sampled for hypothesis testing.
> 7. **Practical Relevance**: We provided examples from healthcare and environmental science to illustrate the real-world application of our method.
> 8. **Related Literature Review**: Noted that due to space constraints, a comprehensive review has been provided in Appendix B in our original version.
> 9. **Experiments Section Complexity**: We've expanded this section for clarity, particularly regarding p-value calculation and the trend of disagreement in synthetic datasets.
>
>
>
> Warm regards,
>
> Authors

---

### Author Response · Authors · 2023-11-18
**Thanks to All Reviewers; Are There Any Other Concerns?**

Dear Reviewer 32x9, Reviewer g7sE, and Reviewer aduf,


**Thank you very much for reviewing our paper. We greatly appreciate all your efforts in helping us!**


We have tried our best to address all your concerns and have revised our paper accordingly. Major revisions include:
- adding more details to the experiment section,
- revising preliminaries,
- emphasizing the absence of latent confounders to clarify the problem setting,
- moving the assumptions from the appendix to the main text to enhance the paper's soundness.


We are thankful for the reviewers' recognition of the *novelty* and *insights* of our paper. Note that this is the first paper to detect the causal direction between instances containing high-dimensional features (such as images) and discrete variables with theoretical proofs. Its *applications* can extend far beyond semi-supervised learning, from *healthcare* and *environmental science* to *economics* (please kindly refer to A7 in our Rebuttal to Reviewer 32x9 for some examples).

We would greatly appreciate it if the reviewers could reconsider the score based on our revised version, the novelty, and the potential impact of our work's broader applications.

Please let us know if there are any further concerns. We are more than happy to address them.


Sincerely,

Authors

---

### Meta-Review · Area_Chair_frHQ · 2023-12-11

**Metareview:**

The paper presents a technique for determining if different parts of a data set have a causal or anti-causal relationship in their generation according to conditional and marginal distributions of the data. The approach checks for correlations between noise levels and the mismatch of distributions learned from the data. The paper received three reviews, with one voting reject, one voting weak accept, and one increasing from reject to weak accept during the discussion phase without modifying the original review to indicate why, including the statement that the paper cannot be recommended for acceptance in its current form. All reviewers only rated the contribution as being fair and concerns were raised about the rigor of the problem setting and soundness of the approach. More empirical support and identifiability results would support the claims being made in lieu of a more theoretically rigorous discussion.

**Justification For Why Not Higher Score:**

Originally had two 3's and one 6. The increase from 3 to 6 wasn't accompanied by a clear indication why the original impression was wrong on the part of the reviewer. The original 6 wasn't a strong argument for acceptance, but seems to be a response to being reviewed for a second time by this reviewer after previous rejection and the fact that some revisions were made. I know that causal reasoning is part of ICLR, but I'm not sure how this fits into the conference. Multiple reviewers seemed hesitant about how theoretically solid the foundations are of the proposed idea.

**Justification For Why Not Lower Score:**

NA

---

### Decision · Program_Chairs · 2024-01-16

Reject